# Structured Progressive Knowledge Activation for LLM-Driven Neural Architecture Search

**Zhen Liu** [* 1 4]  **Yuhan Liu** [* 2]  **Jinjun Wang** [1]  **Wei Song** [3]  **Jianyi Liu** [1]  **Jingwen Fu** [4]

## Abstract

This paper focuses on a key challenge in Neural Architecture Search (NAS): integrating established architectural knowledge while exploring new designs under expensive evaluations. Large language models (LLMs) are a promising assistant for NAS because they can translate rich architectural and coding priors into executable code edits. However, in practice, seemingly local revisions often propagate into non-local behavioral and performance shifts because a single edit can inadvertently couple multiple interacting functional factors, a phenomenon we refer to as functional entanglement. To make LLM knowledge usable under such entanglement, we propose Structured Progressive Knowledge Activation (SPARK), which activates relevant priors by explicitly selecting the functional factor to modify and conditioning the edit on that factor. This factor-conditioned editing reduces entangled side effects and yields more targeted, reliable architecture modifications. On CLRS-DFS, SPARK achieves a $28.1\times$ sample-efficient architecture evolution speedup and yields a 22.9% relative improvement in OOD accuracy. Our code is available at `https://github.com/AIM-ResearchLab/SPARK`.

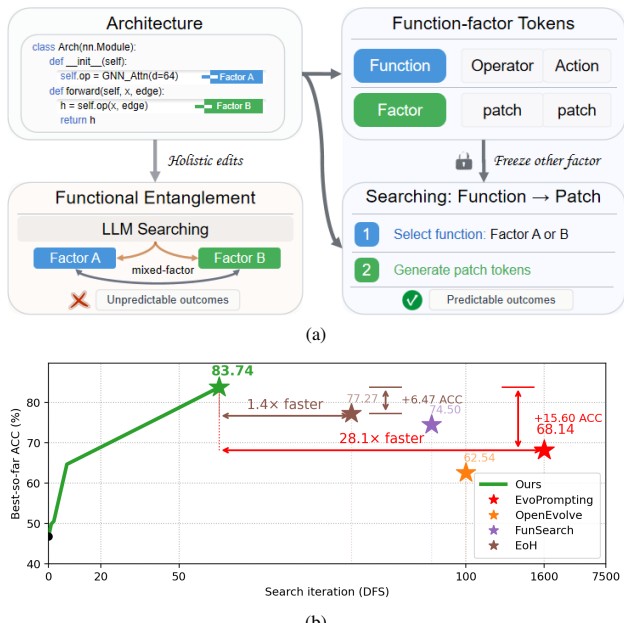

*Figure 1.* Motivation for structure-guided editing in LLM-driven NAS. (a) Free-form edits entangle functional factors and often break interfaces, while factor-scoped tokens enable where-to-how edits to isolate the intervention target. (b) CLRS-DFS results show faster progress and higher OOD accuracy with fewer search iterations.

## 1. Introduction

This paper studies a central tension in Neural Architecture Search (NAS): integrating established architectural knowledge while exploring new designs under expensive evalua-

*Equal contribution [1]State Key Laboratory of Human-Machine Hybrid Augmented Intelligence, Institute of Artificial Intelligence and Robotics, Xi'an Jiaotong University. [2]MiLM Plus, Xiaomi Inc. [3]North China University of Technology, Beijing. [4]Zhongguancun Academy, Beijing, China. Correspondence to: Jingwen Fu <fujingwen@bza.edu.cn>.

*Proceedings of the $43^{rd}$ International Conference on Machine Learning*, Seoul, South Korea. PMLR 306, 2026. Copyright 2026 by the author(s).

tions. This tension is exacerbated by the enormous search space of neural architectures and the high cost of training and evaluation, which makes naive exploration impractical (Zoph & Le, 2017; Real et al., 2017; Liu et al., 2019). LLM-driven NAS is a promising direction because it brings this balance into executable code space: a language model can propose or revise architecture programs by drawing on rich coding and architectural priors, while feedback from training and evaluation guides subsequent iterations (Chen et al., 2023). Yet, in practice, these priors often fail to translate into reliable and controllable edits, becoming a key bottleneck for effective search.

A primary reason is that architecture programs encode multiple interacting functional decisions whose dependencies are largely implicit in code. As illustrated in Fig. 1a (left), "Holistic edits" in free-form "LLM Searching" often update

"Factor A" and "Factor B" together (a "mixed-factor" revision); we call this failure mode functional entanglement. In our setting, "Factor A" corresponds to the Operator factor (which computation module is used), and "Factor B" corresponds to the Action factor (how the operator is invoked and wired under interface and shape constraints). Both factors are functional because together they specify the executable computation graph that determines model behavior. When a single revision entangles these factors, even seemingly local code changes can propagate into non-local behavior and performance shifts and can also break executability (e.g., by violating interface or shape consistency), yielding "Unpredictable outcomes."

These observations suggest that the bottleneck in LLM-driven NAS is not the absence of architectural knowledge, but the lack of a mechanism that turns LLM priors into controllable edits under functional entanglement. To make priors actionable, revisions should be factor-isolated under functional entanglement, matching the constraint in Fig. 1a (right) ("Freeze other factor"). We therefore introduce "Function-factor Tokens" that expose functional factors as explicit, selectable choices, instantiated as Operator and Action, each paired with a dedicated patch slot for localized code changes. This representation supports the structured editing primitive "Searching: Function→Patch": first "Select function: Factor A or B" (i.e., select Operator or Action), then "Generate patch tokens" to realize an executable update conditioned on the selected factor, which promotes more "Predictable outcomes" by isolating the edited factor.

Guided by this principle, we propose Structured Progressive Knowledge Activation (SPARK), a structure-guided editing operator for LLM-driven NAS. SPARK turns free-form rewriting into "Searching: Function→Patch" by decomposing each revision into the two explicit steps in Fig. 1a: "Select function: Factor A or B" and "Generate patch tokens," while enforcing "Freeze other factor" through factor-conditioned patching. By making the intervention factor explicit and restricting each step to one factor at a time, SPARK reduces cross-factor interference between Operator and Action, mitigates entangled side effects, and improves the executability and reliability of generated candidates.

We evaluate SPARK on the CLRS algorithmic reasoning benchmark (Veličković et al., 2022), which instantiates NAS as program-structured architecture tasks under a standardized training and evaluation protocol. On the representative 10-task CLRS subset reported in the main text, SPARK achieves 83.92% mean OOD accuracy with comparable model-level compute measured by MACs relative to the CLRS reference. To avoid task-selection bias, we further report the full 30-task CLRS comparison in Appendix A.1, where SPARK achieves 83.91% mean OOD accuracy, outperforming EvoPrompting (80.89%) and the CLRS reference (75.98%). Beyond aggregate accuracy, we study search dynamics and reliability by tracking best-so-far progress and executability, relating improvements to higher executability and reduced Operator–Action interference.

**Contributions.** (1) We identify functional entanglement as a systematic failure mode of "Holistic edits" in free-form "LLM Searching": single revisions become "mixed-factor" ("Factor A" and "Factor B" jointly updated), leading to "Unpredictable outcomes" including non-local behavior/performance shifts and frequent executability violations. (2) We introduce SPARK, which implements factor-isolated editing via factor selection and factor-conditioned patch generation, operationalized as "Searching: Function→Patch" with "Function-factor Tokens" ("Select function: Factor A or B," "Generate patch tokens," and "Freeze other factor"). (3) On CLRS program-structured architecture tasks, SPARK achieves 83.74% OOD accuracy on DFS with 57 evaluated candidates, yielding a $28.1\times$ sample-efficient architecture evolution speedup over EvoPrompting; it further attains 83.92% mean OOD accuracy on the representative 10-task main-table subset and 83.91% mean OOD accuracy on the full 30-task CLRS suite, with comparable model-level MACs in the main-table comparison, and we analyze search dynamics via best-so-far progress and executability.

## 2. Related Work

**NAS.** Neural architecture search (NAS) automates model design by exploring an architecture space under a validation objective. Representative paradigms include reinforcement-learning controllers (Zoph & Le, 2017), evolutionary search with explicit variation operators (Real et al., 2019), and continuous relaxations for differentiable search (Liu et al., 2019). To reduce evaluation cost, one-shot and weight-sharing methods amortize training by reusing supernet weights (Pham et al., 2018), while standardized benchmarks improve reproducibility and fair comparison (Dong & Yang, 2020). A common practice in these pipelines is to specify a structure-aware search operator (e.g., mutate a particular edge/operator, perturb a cell, or update a relaxed parameterization), which makes the effects of each step easier to control and attribute. This property becomes fragile in LLM-driven NAS, where the search step is implemented as free-form code rewriting rather than an explicit operator, making functional roles implicit and edits harder to control.

**LLM-driven NAS.** Recent work explores LLMs for code-level NAS as architecture generators/refiners and as adaptive mutation operators under evaluation feedback (Chen et al., 2023; Zheng et al., 2023; Nasir et al., 2024; Wu et al., 2023). Beyond direct edits, LLM priors have also been leveraged via distilled design principles for warm-starting (Zhou et al.,

2025), by coupling reflection with zero-cost proxies (Ji et al., 2025), and by coordinating multiple LLM roles during search (Li et al., 2025). These methods demonstrate that LLM priors can guide exploration, but the per-iteration update is often treated as a single holistic rewrite or proposal. As a result, interacting functional decisions that are not explicitly surfaced in code can be coupled within one revision, such as changing an operator together with its action-level wiring and related interface/shape constraints. In our setting, this holistic-edit regime can induce Functional Entanglement, where Operator and Action factors are jointly modified, making the effect of an edit harder to attribute and reducing how precisely architectural priors are activated and reused across rounds.

**Iterative LLM optimization and structured editing.** A parallel literature studies iterative refinement loops that improve outputs using self-feedback or external signals (Madaan et al., 2023; Shinn et al., 2023; Yang et al., 2024; Fernando et al., 2024). LLMs have also been integrated with evolutionary computation and program search as mutation/crossover operators and proposal mechanisms (Lehman et al., 2022; Meyerson & Miikkulainen, 2023; Šurina et al., 2025; Hemberg & O'Reilly, 2024), with surveys systematizing this emerging interplay (Wu et al., 2025). Separately, structure-aware code generation/editing constrains outputs to grammars or ASTs (Yin & Neubig, 2017; Rabinovich et al., 2017), and program repair restores compilability/executability from erroneous code (Gupta et al., 2017). These lines motivate validity constraints, but they do not directly address the NAS-specific failure mode we study: multi-round evolution in architecture code where implicit functional roles allow a single free-form edit to couple Operator-level changes with Action-level wiring, leading to Functional Entanglement. Our work complements them by making Operator/Action factors explicit via Function-factor Tokens and enforcing factor-isolated edits at the iteration level through Searching: Function→Patch, together with feasibility checks, to reduce cross-factor interference in LLM-driven NAS.

## 3. Method

### 3.1. Problem Setup

We study LLM-driven multi-round neural architecture search (NAS) in code space (Chen et al., 2023), where each candidate architecture is represented as executable code $a \in \mathcal{A}$ implementing a neural algorithmic architecture under a fixed training and evaluation pipeline. Given a task $\mathcal{T}$ and an evaluation budget $B$ (the maximum number of candidates that can be fully trained and evaluated) (Zoph & Le, 2017; Real et al., 2017; Liu et al., 2019), each executable

candidate is scored by a black-box evaluator

$$\mathrm{Eval}_{\mathcal{T}}(a) \to \mathbf{m}(a) = \big(f(a), \phi(a)\big), \quad (1)$$

where $f(a)$ is the primary performance metric (e.g., OOD accuracy under the task protocol (Veličković et al., 2022)) and $\phi(a)$ is a descriptor vector derived from resource signals (e.g., MACs and parameter count) (Mouret & Clune, 2015). We keep the training and evaluation pipeline fixed across candidates so that improvements are attributable to architecture edits rather than changes in optimization or data processing.

We distinguish between proposed candidates and evaluated candidates. A proposed candidate is produced by the LLM editor. Only candidates that are executable and pass feasibility checks enter $\mathrm{Eval}_{\mathcal{T}}(\cdot)$, become evaluated candidates, and count toward the evaluation budget $B$. Unless otherwise specified, the seed architecture $a_0$ is evaluated and counts toward $B$.

### 3.2. Structured Progressive Knowledge Activation (SPARK)

We propose Structured Progressive Knowledge Activation (SPARK), a factor-conditioned editing method that disentangles intervention target selection from factor-conditioned refinement. At iteration $t$, we maintain an evolution context $\mathcal{H}_t$ summarizing the parent candidate, recent outcomes, and optional inspiration candidates sampled from an archive. SPARK progressively activates structured knowledge from $\mathcal{H}_t$ in three stages: (i) Architecture Scope Router (ASR) selects the functional factor to edit, (ii) Refinement Compass (RC) converts recent search signals into a factor-conditioned refinement directive, and (iii) Scoped Architecture Refiner (SAR) applies a single-shot code edit under the selected factor and directive. After this first definition, we use ASR, RC, and SAR as abbreviations.

**Factor-conditioned evolution step.** Each SPARK step factorizes as

$$\begin{aligned} f_t &\leftarrow \mathrm{ASR}(a_t, \mathcal{H}_t), \\ d_t &\leftarrow \mathrm{RC}(a_t, f_t, \mathcal{H}_t), \quad (2) \\ \tilde{a}_{t+1} &\leftarrow \mathrm{SAR}(a_t, f_t, d_t, \mathcal{H}_t), \end{aligned}$$

where $f_t$ is a discrete factor indicating the intervention target, $d_t$ is a refinement directive, and $\tilde{a}_{t+1}$ is the proposed offspring program. Intuitively, ASR answers what to intervene on (target selection), while RC+SAR specify how to revise under the chosen factor (factor-conditioned refinement).

**LLM roles and decoding.** SPARK uses a single base LLM with role-specific prompts for routing/directive/editing (denoted as $\mathrm{LLM}_{\mathrm{route}}$, $\mathrm{LLM}_{\mathrm{dir}}$, and $\mathrm{LLM}_{\mathrm{edit}}$ for clarity).

All LLM calls use the same decoding hyperparameters (temperature $= 0.7$, other settings fixed) and we report results over multiple random seeds.

**CLRS instantiation: factorization into OPERATOR and ACTION.** In our CLRS-based architecture programs, the editable design naturally separates into two intrinsic factors

$$S = \{\text{OPERATOR, ACTION}\}. \tag{3}$$

OPERATOR modifies operator choices and parameterizations (e.g., projections and gating primitives), while ACTION modifies how operators are invoked and composed in the forward computation (e.g., message construction/wiring, control flow, and masking action). In the implementation, these map to two disjoint code regions in the program, while non-architectural scaffolding (public interfaces, training pipeline hooks, and task I/O) remains frozen.

**Factor-scoped tokenization (factor tokens + region tags).** We annotate each candidate program with two disjoint code regions corresponding to OPERATOR and ACTION. Concretely, we wrap each region with fixed boundary tags (region markers) and require the LLM editor to output a complete updated program while preserving the tags. The router (ASR) emits a single factor token indicating $f_t \in S$; the factor token is restricted to $\{\text{OPERATOR, ACTION}\}$ and is parsed deterministically. This design makes the edit target explicit (via factor tokens) and enables factor-locality enforcement (via region tags + feasibility checks).

### 3.3. Factor-Respecting Feasibility

We define the factor-local feasible set for a parent $a_t$ and factor $f_t$ as

$$\mathcal{F}(a_t, f_t) \triangleq \Big\{ a \in \mathcal{A} \ \Big| \ \text{Edit}(a, a_t) \subseteq \mathcal{R}_{f_t}, \\ a \models \mathcal{E}_{f_t} \wedge \mathcal{C} \Big\}, \tag{4}$$

where $\mathcal{R}_{f_t}$ is the editable code region for factor $f_t$, $\mathcal{E}_{f_t}$ denotes template/schema constraints (preserve boundary tags, edit only the designated region, and output a syntactically complete program), and $\mathcal{C}$ denotes semantic constraints shared across factors (interface invariants and tensor-shape/masking requirements). Importantly, $\mathcal{C}$ is factor-invariant.

**Operationalizing edits and regions.** We compute $\text{Edit}(a, a_t)$ as a line-based diff over a normalized code representation. Normalization is purely formatting-level: we (i) normalize line endings to LF, and (ii) strip trailing whitespaces on each line; no semantic code transformation (e.g., comment removal, import reordering) is performed. We

**Algorithm 1** Structured Progressive Knowledge Activation (SPARK)

---
1: **Input:** Initial architecture $a_0$; factors $S = \{\text{OPERATOR, ACTION}\}$
2: **Input:** Evaluation budget $B$; RC stagnation window $k = 3$; proposal window $k' = 10$
3: **Input:** Router retries $R_{\text{asr}}$; archive discretizer $b(\cdot)$; migration period $P$
4: $(f, \phi) \leftarrow \text{Eval}_\mathcal{T}(a_0)$; initialize islanded MAP-Elites archive $\mathcal{D}$ with $a_0$
5: $n_{\text{eval}} \leftarrow 1$
6: Initialize a FIFO buffer $\mathcal{Q}_{\text{prop}}$ to store recent proposal feasibility outcomes (size $k'$)
7: **while** $n_{\text{eval}} < B$ **do**
8:     Sample parent $a_t$ (and inspirations) from $\mathcal{D}$
9:     Build context $\mathcal{H}_t$ (parent, recent outcomes, inspirations, $\mathcal{Q}_{\text{prop}}$ summary)
10:     $f_t \leftarrow \text{ASR}(a_t, \mathcal{H}_t, S, R_{\text{asr}})$     // target selection
11:     $d_t \leftarrow \text{RC}(a_t, f_t, \mathcal{H}_t, k, k')$   // directive under factor
12:     $\tilde{a} \leftarrow \text{SAR}(a_t, f_t, d_t, \mathcal{H}_t)$     // single-shot edit
13:     **if** $\tilde{a} = \text{FAIL}$ **or** $\tilde{a} \notin \mathcal{F}(a_t, f_t)$ **then**
14:         Push FAIL (with failure type) to $\mathcal{Q}_{\text{prop}}$; **continue**
15:     **end if**
16:     Push PASS to $\mathcal{Q}_{\text{prop}}$
17:     $(f, \phi) \leftarrow \text{Eval}_\mathcal{T}(\tilde{a})$
18:     $n_{\text{eval}} \leftarrow n_{\text{eval}} + 1$
19:     $c \leftarrow b(\phi)$; update archive cell $c$ if improved (tie-break by lower MACs)
20:     Migrate elites every $P$ iterations across islands
21: **end while**
22: **return** best archived architecture(s) in $\mathcal{D}$
---

map changed line ranges to regions using the fixed boundary tags that delimit $\mathcal{R}_{\text{OPERATOR}}$ and $\mathcal{R}_{\text{ACTION}}$. A proposal is accepted only if all modified lines fall within the selected region $\mathcal{R}_{f_t}$ and the entire non-selected region text (between boundary tags) is byte-identical to the parent after normalization.

**Feasibility pipeline.** In practice, we enforce $\tilde{a}_{t+1} \in \mathcal{F}(a_t, f_t)$ using a lightweight feasibility pipeline: (1) syntax check (parse/import of the generated program), (2) interface check (frozen signatures and required symbols are present and unchanged), and (3) semantic check (a forward-only pass under the CLRS input specification to validate tensor shapes and masking invariants, including correct handling of adjacency-defined non-edges). This pipeline is orders of magnitude cheaper than full training. Candidates failing any check are rejected before evaluation and do not count toward the evaluation budget; remaining training instabilities (if any) are reflected by $\text{Eval}_\mathcal{T}(\cdot)$.

### 3.4. Search Backbone and Overall Procedure

**Backbone-agnostic interface.** SPARK consists of a factor-conditioned editing operator (Sec. 3.2) and feasibility enforcement (Sec. 3.3). It can be integrated into any multi-round search procedure that (i) selects a parent archi-

**Algorithm 2** SPARK Modules: ASR, RC, and SAR

---

1:    **Procedure** ASR($a_t, \mathcal{H}_t, \mathcal{S}, R_{asr}$)
2:      **for** $r = 1$ **to** $R_{asr}$ **do**
3:         $y \leftarrow \text{LLM}_{route}(\text{RoutePrompt}(a_t, \mathcal{H}_t))$
4:         $f \leftarrow \text{ParseFactor}(y)$
5:         **if** $f \in \mathcal{S}$ **then return** $f$ **end if**
6:      **end for**
7:      **return** ACTION

8:    **Procedure** RC($a_t, f_t, \mathcal{H}_t, k, k'$)
9:      **Definition (evaluated).** An evaluated candidate is one that passes feasibility checks and enters $\text{Eval}_\mathcal{T}(\cdot)$ (thus counted toward the budget).
10:     Compute stagnation over the last $k$ evaluated candidates in $\mathcal{H}_t$ (e.g., best-so-far improvement).
11:     Compute feasibility failure rate and dominant failure types over the last $k'$ proposals using $\mathcal{Q}_{prop}$.
12:     $d \leftarrow \text{LLM}_{dir}(\text{DirectivePrompt}(a_t, f_t, \text{signals}))$
13:     **return** Sanitize($d$)

14:   **Procedure** SAR($a_t, f_t, d_t, \mathcal{H}_t$)
15:     $\tilde{a} \leftarrow \text{LLM}_{edit}(\text{EditPrompt}(a_t, f_t, d_t, \mathcal{H}_t))$
16:     **if** HasRegionTags($\tilde{a}$) **then return** $\tilde{a}$ **end if**
17:     **return** FAIL

---

tecture (optionally with inspirations) and (ii) evaluates the edited candidate to provide fitness feedback. The search backbone only affects the sampling policy across rounds; our contribution is the SPARK operator itself.

**Backbone used in experiments.** In this paper, we instantiate the search loop with an archive-based evolutionary backbone (OpenEvolve) (Sharma, 2025), which maintains an archive of elites and replaces an incumbent when a new candidate achieves higher fitness. We use the default OpenEvolve settings unless otherwise specified.

Algorithm 1 summarizes the overall pipeline under this instantiation, and Algorithm 2 details the three SPARK modules.

## 4. Experiments and Results

We evaluate **Structured Progressive Knowledge Activation (SPARK)** on the CLRS Algorithmic Reasoning Benchmark (Veličković et al., 2022), following the same benchmark family considered by EvoPrompting (Chen et al., 2023). Our primary controlled comparison is between **OpenEvolve (OE)** and **SPARK** under an identical search backbone and an identical LLM editor, so that differences are attributable to SPARK's where-then-how, factor-scoped evolution operator rather than the underlying evolutionary infrastructure or language model.

Unless otherwise specified, the evolutionary loop is built on OpenEvolve (Sharma, 2025), and the generator/editor is DeepSeek-R1-0528 (DeepSeek, 2025) accessed via an

OpenAI-compatible API endpoint. We treat the LLM as a black-box code editor: we use a single prompt template throughout, with no prompt-tuning, no multi-prompt rounds, and no model-side adaptation. Search is initialized from the canonical CLRS reference implementation, and candidates are obtained by iterative edits of this single seed program (without changing the underlying model family), which reduces confounding human priors from multi-seed initialization and isolates the effect of SPARK's factor-scoped edits.

To encourage diversity, we use an island-based population (5 islands, population size 100) with an in-memory archive of size 100. We discretize archive cells using descriptors over $\{\texttt{ood\_acc}, \texttt{macs}\}$. At each iteration, the prompt includes both exploitation and exploration context by inserting the top-performing programs (5) and a set of diverse programs (5) sampled from the archive. We enable cascade evaluation to filter clearly poor candidates before full evaluation (threshold $-100$), and use robust request handling (timeouts and retries) to reduce interruptions during long runs.

**Budget Accounting.** We distinguish between proposal attempts and evaluated candidates. A proposal attempt refers to an LLM-generated edit, indexed by the search iteration. A candidate counts as evaluated only if it is executable, passes feasibility checks, and is fully trained and scored under the CLRS protocol. Non-executable proposals are rejected before evaluation and therefore do not count toward the evaluation budget. Accordingly, our reliability curves in Fig. 2 report metrics over a fixed 100-attempt trajectory, while evaluation efficiency is reported in terms of evaluated candidates.

**Search-and-Transfer Protocol.** For computational comparability, we perform architecture evolution only on the CLRS `dfs` task and then transfer the best-found architecture to the remaining CLRS tasks without task-specific evolution. In contrast, EvoPrompting performs evolution across multiple tasks. Under this setting, performance on non-`dfs` tasks reflects the cross-task generalization of the evolved architecture and the robustness of SPARK's search dynamics.

All candidates are trained and evaluated under the standard CLRS protocol (Veličković et al., 2022). We report OOD accuracy and MACs, and analyze search efficiency and reliability by tracking best-so-far progress, entanglement, and executability as a function of proposal attempts and evaluated candidates.

**Search Cost and Cross-Benchmark Evaluation.** To further examine whether the proposed factor-conditioned editing mechanism is specific to CLRS, we evaluate SPARK on five non-CLRS benchmarks spanning classification, text classification, reinforcement learning, protein-

*Table 1.* Representative 10-task comparison of model size (MACs) and OOD accuracy on CLRS (Veličković et al., 2022). We report results for the CLRS reference architecture (CLRS), OpenEvolve (OE) (Sharma, 2025), EvoPrompting (EP) (Chen et al., 2023), FunSearch (FS) (Romera-Paredes et al., 2024), EoH (Liu et al., 2024), and our method (Ours; i.e., SPARK), including a single-iteration variant (Ours-1it). Our model size is measured by MACs and can be viewed as the CLRS reference model compute plus a small, fixed overhead introduced by the SPARK editing interface, which is constant across tasks. Therefore, gains in OOD accuracy mainly stem from improved search dynamics enabled by factor-isolated edits (operator vs. action), rather than increased model capacity or architecture-level MACs. **Abbreviations:** AP=Articulation Points, QS=Quickselect, SCC=Strongly Connected Components, AS=Activity Selector, FMS=Find Maximum Subarray (Kadane), Dij=Dijkstra, LCS=LCS Length, Min=Minimum, TS=Topological Sort.

| Metric / Method | | DFS | AP | QS | SCC | AS | FMS | Dij | LCS | Min | TS | Avg. |
|---|---|---|---|---|---|---|---|---|---|---|---|---|
| | | | | | | CLRS task (abbr.) | | | | | | |
| MACs (K) | CLRS | 661 | 532 | 396 | 663 | 262 | 265 | 526 | 270 | 261 | 660 | 450 |
| | OE | 693 | 563 | 427 | 695 | 294 | 296 | 557 | 302 | 293 | 692 | 481 |
| | EP | 660 | 498 | 377 | 707 | 262 | 261 | 525 | 270 | 260 | 660 | 448 |
| | FS | 680 | 551 | 415 | 683 | 282 | 284 | 545 | 290 | 281 | 679 | 469 |
| | EoH | 675 | 546 | 410 | 677 | 276 | 278 | 540 | 284 | 275 | 674 | 463 |
| | Ours-1it | 670 | 541 | 405 | 673 | 271 | 274 | 535 | 280 | 271 | 669 | 459 |
| | Ours | 664 | 535 | 399 | 666 | 265 | 268 | 529 | 273 | 264 | 663 | 453 |
| OOD acc.(%) | CLRS | 46.78 | 88.32 | 0.47 | 43.43 | 95.18 | 76.36 | 96.05 | 80.51 | 97.78 | 87.27 | 71.22 |
| | OE | 32.54 | 61.32 | 3.71 | 45.21 | 95.43 | 77.05 | 95.85 | 87.53 | 98.10 | 86.31 | 68.30 |
| | EP | 68.14 | 93.46 | 0.79 | 41.86 | 95.05 | 75.35 | 97.30 | 85.75 | 98.40 | 88.12 | 74.42 |
| | FS | 74.50 | 94.07 | 1.49 | 62.08 | 95.61 | 77.75 | 96.84 | 82.93 | 97.91 | 92.88 | 77.61 |
| | EoH | 77.27 | 95.03 | 4.25 | 65.90 | 96.22 | 80.51 | 97.44 | 84.56 | 98.14 | 95.00 | 79.43 |
| | Ours-1it | 45.60 | 73.04 | 9.57 | 51.56 | 97.54 | 79.79 | 97.80 | 86.89 | 99.02 | 84.94 | 72.58 |
| | Ours | **83.74** | **97.90** | **10.69** | **77.34** | **98.03** | **85.64** | **99.22** | **87.60** | **99.07** | **99.95** | **83.92** |

*Table 2.* Summary of additional non-CLRS benchmark results on MNIST-1D (Greydanus & Kobak, 2020), 20News (Mitchell, 1999), CartPole-v1 (Gavenski et al., 2024), Mice Protein Expression (Higuera et al., 2015), and Breast Cancer Wisconsin (Wolberg et al., 1993). Values denote accuracy except for CartPole-v1, where the score is mean test return. For Mice Protein Expression, this summary reports accuracy only; loss is reported in Appendix A.5.

| Benchmark | OpenEvolve | Ours |
|---|---|---|
| MNIST-1D | 70.25 | **95.99** |
| 20News | 57.10 | **59.13** |
| CartPole-v1 | 377.55 | **489.00** |
| Mice Protein Expr. | **61.78** | **61.78** |
| BCW | 96.90 | **97.04** |

expression classification, and medical diagnosis: MNIST-1D (Greydanus & Kobak, 2020), 20News (Mitchell, 1999), CartPole-v1 (Gavenski et al., 2024), Mice Protein Expression (Higuera et al., 2015), and Breast Cancer Wisconsin (Wolberg et al., 1993). Table 2 compares SPARK with the vanilla OpenEvolve baseline under the same high-level search interface, while Appendix A.5 provides detailed results for all baselines.

## 4.1. CLRS Architecture

We evaluate SPARK on the CLRS algorithmic reasoning benchmark (Veličković et al., 2022), a standardized suite designed to test whether neural networks can learn algorithmic

reasoning across a diverse set of classical algorithms drawn from standard curricula (Cormen et al., 2009). CLRS is particularly suitable for studying multi-round LLM-driven evolution because it requires systematic generalization beyond pattern matching: models must infer and execute algorithmic computation over structured inputs, rather than relying on superficial correlations or short-horizon heuristics. Following prior work, we view success on CLRS as evidence of improved algorithmic alignment—the ability to solve problems by reasoning in a manner consistent with the underlying computation graph of the target algorithm (Xu et al., 2020).

**Benchmark structure.** In CLRS, each algorithmic instance is represented as a graph-structured input, and the algorithm is expressed as a trajectory of operations over this graph (e.g., node/edge updates and readouts) (Veličković et al., 2022). This formulation naturally supports message-passing models and allows evaluation across a broad set of algorithms under a shared interface. Prior studies have shown that graph neural networks can process such trajectories effectively, and that carefully designed message-passing architectures can provide strong performance across many CLRS tasks (Ibarz et al., 2022; Gilmer et al., 2017).

**Architecture instantiation and search target.** To isolate the effect of multi-round evolution from changes in model family, we operate directly on the original CLRS ref-

erence architecture provided by the benchmark (Veličković et al., 2022). That is, our search starts from the official reference implementation and iteratively edits its Python function definition. As illustrated in Fig. 1, this code-based representation exposes two intrinsic, disjoint architecture factors: (i) **OPERATOR** (module parameterization/structure definitions, e.g., components defined in `__init__`), and (ii) **ACTION** (how operators are composed, reused, masked, and routed in `forward`). SPARK implements a where-then-how evolution operator that factorizes each update into: where = discrete scope selection over {`OPERATOR`, `ACTION`} (ASR), followed by how = within-scope refinement under an explicit directive (RC+SAR). This separation is designed to reduce cross-factor interference and make improvements more composable across iterations.

## 4.2. Main Results on CLRS

**Evaluation protocol.** We use a search-and-transfer evaluation protocol. For each method, we first run LLM-driven evolutionary search on the CLRS `dfs` task for 100 proposal attempts under the same search budget and evaluation harness, and measure search performance using the CLRS OOD split. We then transfer the incumbent best architecture discovered on `dfs` to nine additional representative CLRS tasks, where it is trained and evaluated from scratch for the compact main-table comparison. We report (i) per-task OOD accuracy and MACs on this representative 10-task subset in Table 1, (ii) full OOD accuracy over all 30 CLRS tasks in Appendix A.1, and (iii) reliability/progress curves over the 100-attempt trajectory in Fig. 2.

**Baselines and Reproducibility.** We compare SPARK with EvoPrompting (Chen et al., 2023), OpenEvolve (Sharma, 2025), FunSearch (Romera-Paredes et al., 2024), and EoH (Liu et al., 2024), as listed in Table 1. Since OpenEvolve and FunSearch do not explicitly report CLRS results, we obtain their numbers by running their released code in our CLRS pipeline, using the same `dfs`-based evolution setting and the same 100-attempt budget as SPARK for a controlled comparison.

**Evaluation-Efficiency Accounting.** On `dfs`, SPARK reaches its best OOD accuracy of 83.74% by proposal attempt 57, corresponding to 57 evaluated candidates in that run. Since this already surpasses EvoPrompting's best DFS result under its 1600-evaluation setting, the evaluated-candidate accounting gives a $1600/57 \approx 28.1\times$ improvement in evaluation efficiency.

Table 1 summarizes OOD accuracy and model size (MACs) on the representative 10-task CLRS subset. Across these listed tasks, SPARK (Ours) achieves the best OOD accuracy, outperforming the CLRS reference architecture as well

as strong LLM-evolution baselines (OE/EP/FS/EoH), indicating that structure-guided, factor-scoped editing supports more reliable multi-round progress under the same proposal budget. The gains are particularly pronounced on graph-reasoning tasks: on DFS we improve from 46.78 (CLRS) / 74.50 (FS) to 83.74 (Ours), and on SCC from 43.43 (CLRS) / 62.08 (FS) to 77.34. We also observe substantial relative gains in regimes where the reference model performs poorly, e.g., QS ($0.47 \rightarrow 10.69$), highlighting the benefit of controlling where to edit before generating how to edit.

The single-iteration variant (Ours-1it) already improves over CLRS on several tasks (e.g., QS: $0.47 \rightarrow 9.57$), but remains clearly behind Ours (QS: 10.69; DFS: 83.74), supporting the necessity of multi-round evolution for sustained gains.

Importantly, these improvements are not explained by capacity/compute scaling. As shown by the MACs in Table 1, Ours stays close to the CLRS reference compute across tasks (e.g., 661K vs. 664K on DFS; 663K vs. 666K on SCC), and is often comparable to or smaller than the baselines. Therefore, the accuracy gains primarily reflect improved search dynamics induced by factor-scoped updates rather than increased compute.

Finally, tasks that require multi-step, compositional reasoning (e.g., DFS, SCC, and TS) benefit most, with TS reaching 99.95 (Ours) compared to 87.27 (CLRS) and 92.88 (FS), consistent with our hypothesis that reducing cross-factor entanglement improves iterative search stability.

Beyond this compact 10-task main-table comparison, Appendix A.1 reports the full 30-task CLRS comparison to address task-selection bias. Averaged over all 30 tasks, SPARK obtains 83.91% mean OOD accuracy, compared with 80.89% for EvoPrompting and 75.98% for the CLRS reference. SPARK improves over the CLRS reference on 21/30 tasks and over EvoPrompting on 19/30 tasks. These full-suite results show that the gains are not limited to the representative subset, while also revealing remaining task-transfer failures such as BFS, Floyd–Warshall, and KMP.

## 4.3. Ablation on Factorized Editing

**Ablation on factorized edits on DFS.** SPARK factorizes each evolution step into ASR (scope selection) and RC+SAR (scope-local refinement). Table 3 reports DFS results under two LLM editors (DeepSeek-R1 and Qwen-Plus). With DeepSeek-R1, using only RC+SAR yields a modest improvement over the CLRS baseline ($46.78 \rightarrow 56.79$, +10.01 pp), while using only ASR brings a larger gain ($46.78 \rightarrow 65.28$, +18.50 pp), suggesting that correctly localizing the intervention scope is more critical than refinement alone. Combining ASR and RC+SAR (SPARK) achieves the best accuracy (83.74, +36.96 pp) with MACs close to the reference (664K vs. 661K), indicating the gains

*Table 3.* DFS ablation (CLRS): ASR vs. RC+SAR under different LLM editors. We compare using only RC+SAR, only ASR, and full SPARK (ASR+RC+SAR) with identical evaluation protocol. OOD is OOD accuracy on DFS over valid (evaluable) proposals; $\Delta$ is the gain (pp) over the CLRS baseline.

| Variant | LLM | MACs↓ | OOD↑ | $\Delta$↑ |
|---------|-----|-------|------|-----------|
| CLRS (base) | DeepSeek-R1 | 661,190 | 46.78 | +0.00 |
| RC+SAR only | DeepSeek-R1 | 727,823 | 56.79 | +10.01 |
| ASR only | DeepSeek-R1 | 693,926 | 65.28 | +18.50 |
| Ours (SPARK) | DeepSeek-R1 | 664,157 | **83.74** | **+36.96** |
| RC+SAR only | Qwen-Plus | 731,000 | 56.00 | +9.22 |
| ASR only | Qwen-Plus | 697,000 | 64.50 | +17.72 |
| Ours (SPARK) | Qwen-Plus | 665,308 | **80.50** | **+33.72** |

are not driven by capacity scaling. We observe the same qualitative trend with Qwen-Plus: RC+SAR only improves to 56.00 (+9.22 pp) and ASR only reaches 64.50 (+17.72 pp), whereas SPARK attains 80.50 (+33.72 pp) with comparable MACs (665K).

Appendix A.4 further reports scope-control baselines that isolate the effect of region tags alone, and provides an illustrative step-level example of how ASR, RC, and SAR interact in one SPARK iteration.

### 4.4. Search Dynamics and Reliability

These ablations also explain the reliability patterns in Fig. 2. ASR explicitly constrains edits to a single scope, which reduces cross-scope entanglement (Fig. 2b) and in turn increases the fraction of proposals that remain executable and evaluable (Fig. 2c). As a result, more of the fixed 100-attempt budget is converted into effective optimization steps, yielding faster and higher best-so-far OOD accuracy (Fig. 2a). RC+SAR alone can refine locally once a good scope is implicitly hit, but without reliable scope localization it is more likely to produce edits whose effects are harder to attribute or that interfere with other architectural factors, which aligns with the higher entanglement and lower validity observed for unconstrained editing. Overall, separating scope selection (ASR) from within-scope refinement (RC+SAR) reduces entanglement, improves executability, and leads to more reliable multi-round progress.

**Why functional entanglement hurts multi-round search.** Multi-round evolution benefits from making small, attributable changes that can compose across iterations. When an edit touches multiple scopes (or leaks into frozen scaffolding), it effectively introduces multiple coupled constraints at once, increasing the chance of violating interface/shape requirements and reducing the fraction of proposals that remain evaluable. A simple way to see this is to model feasibility as a per-scope event: if changing one scope preserves

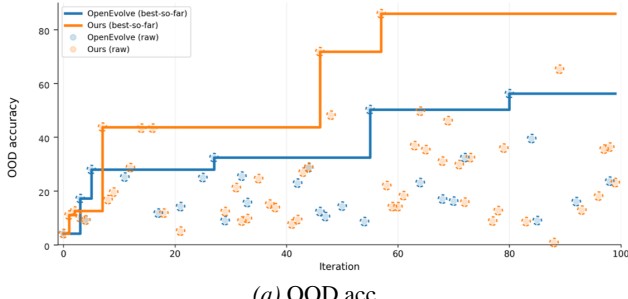

*(a)* OOD acc.

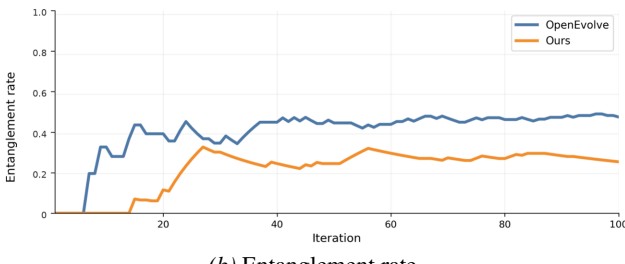

*(b)* Entanglement rate.

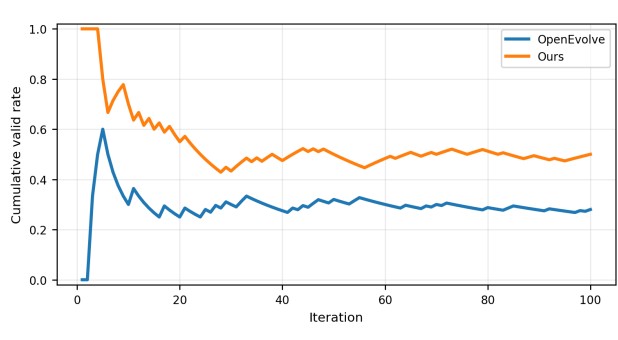

*(c)* Cumulative valid rate.

*Figure 2.* SPARK vs. OpenEvolve under a 100-attempt budget: (a) best-so-far OOD accuracy over valid (evaluable) candidates, (b) entanglement rate measured by non-factor-local (cross-scope) edits, and (c) cumulative valid rate (executability). Together, lower entanglement increases the fraction of evaluable proposals and accelerates best-so-far accuracy gains.

executability with probability $p_v$, then an entangled edit affecting $k$ scopes has feasibility approximately $p_v^k$, which drops quickly as $k$ increases. Moreover, coupled multi-scope changes confound credit assignment: even if one local modification is beneficial, side effects from other touched regions can cancel the gain or destabilize training, reducing the probability that improvements are retained and amplified across rounds. This motivates enforcing scope-local edits (via ASR) before applying within-scope refinement (via RC+SAR), which directly targets lower entanglement, higher validity, and more reliable best-so-far progress as observed in Fig. 2.

**Functional entanglement rate.** Fig. 2b reports the entanglement rate during search. We define an edit as entangled if its code diff is not factor-local: it modifies both functional

factor regions (Operator and Action) in the same revision, or touches any frozen scaffolding outside these regions (e.g., interfaces/I/O hooks). Operationally, we compute a normalized line-based diff between a parent and its offspring and check which predefined code regions the changed lines fall into; the exact criterion and region definitions are provided in Appendix. Compared with OpenEvolve, SPARK yields a consistently lower entanglement rate, indicating fewer mixed-factor or scaffolding-touching revisions

**Reliability of proposal generation.** Fig. 2c evaluates reliability under a fixed 100-attempt budget via the cumulative valid rate, where a proposal is valid if it compiles/runs and passes feasibility checks to enter scoring. SPARK maintains a substantially higher valid rate throughout the trajectory (stabilizing around 0.5–0.6 by iteration 100) than OpenEvolve (around 0.25–0.3), meaning a larger fraction of attempts become evaluable candidates. This improvement aligns with the lower entanglement rate: cross-factor or scaffolding-touching edits are more likely to violate interface/shape constraints and fail before evaluation.

**Search progress under a fixed attempt budget.** Fig. 2a plots best-so-far OOD accuracy over valid candidates. Under the same 100 attempts, SPARK improves faster and reaches 83.74 at iteration 57, while OpenEvolve peaks later (around iteration 80). Taken together, Figs. 2a–2c suggest a tight coupling between entanglement, executability, and progress: reducing entanglement tends to increase the fraction of evaluable proposals, which makes a larger portion of the fixed attempt budget effective for improving best-so-far accuracy.

## 5. Limitations

SPARK improves the controllability and reliability of LLM-driven architecture evolution, but several limitations remain. First, an architecture evolved on `dfs` does not uniformly transfer to all CLRS tasks; for example, BFS, Floyd–Warshall, and KMP remain challenging in the full 30-task evaluation. Second, SPARK introduces additional LLM calls for routing, directive generation, and scoped refinement, so its search-time cost is higher than single-call rewriting baselines even though the final model-level MACs remain comparable. Third, the current factorization uses two manually defined functional regions, OPERATOR and ACTION; extending SPARK to finer-grained or automatically discovered functional factors is an important direction for future work.

## 6. Conclusion

We study LLM-driven neural architecture search as multi-round evolution over executable programs and identify cross-factor functional entanglement as a key reliability bottleneck. We propose SPARK, which enables factor-isolated editing via factor selection and factor-conditioned patch generation, instantiated on CLRS with OPERATOR/ACTION factors and feasibility checks.

On the full 30-task CLRS suite, SPARK achieves 83.91% mean OOD accuracy, improving over the CLRS reference on 21/30 tasks and over EvoPrompting on 19/30 tasks. On DFS, SPARK reaches 83.74% OOD accuracy with 57 evaluated candidates, yielding a $28.1\times$ speedup under our evaluation accounting. Cost accounting shows that SPARK trades additional ASR–RC–SAR LLM calls for higher final architecture quality, achieving the best mean OOD accuracy with comparable model-level MACs. Beyond CLRS, SPARK improves or matches the OpenEvolve baseline across five additional benchmarks, with a notable gain on MNIST-1D (70.25→95.99), suggesting that factor-conditioned editing can transfer across diverse task interfaces.

## Impact Statement

This paper presents work whose goal is to advance the reliability of LLM-driven architecture evolution and neural architecture search. While the techniques studied here may improve the efficiency and reproducibility of model development, we do not anticipate direct negative societal impacts beyond those commonly associated with general-purpose machine learning research.

## Acknowledgement

This work is supported by the Zhongguancun Academy, (Grant No. XTS0070). This work is also supported by the National Key R&D Program of China (2024YFB3309303). Jingwen Fu and Zhen Liu are partially supported by State Key Laboratory of Human-Machine Hybrid Augmented Intelligence, Xi'an Jiaotong University, No. HMHAI-KF202504 and Beijing Natural Science Foundation (No. L253018).

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

# A. Supplementary Material for Structured Progressive Knowledge Activation

## A.1. Overall Comparison with EvoPrompting on the Full 30-Task CLRS Suite

Table A.1 reports the full 30-task OOD accuracy comparison among the CLRS reference, EvoPrompting, and SPARK. Averaged over all 30 tasks, the CLRS reference, EvoPrompting, and SPARK obtain 75.98%, 80.89%, and 83.91% mean OOD accuracy, respectively. SPARK improves over the CLRS reference on 21/30 tasks and over EvoPrompting on 19/30 tasks, indicating that the representative main-table results are consistent with the full-suite trend.

## A.2. Search Dynamics under Structure-Guided Evolution

Fig. A.1 characterizes the search behavior of our Structured Progressive Knowledge Activation (SPARK) over the 100 generated candidates on CLRS, aggregated across five random seeds. Recall that SPARK factorizes each evolution step into where-to-edit and how-to-refine decisions: the Architecture Scope Router (ASR) selects an explicit scope (e.g., operator-level vs. action), the Refinement Compass (RC) specifies scope-local refinement directives, and the Scoped Architecture Refiner (SAR) applies edits only within the chosen scope under feasible-set refinement constraints (syntax/interface/shape feasibility), preventing cross-factor entanglement.

**Progress (best-so-far OOD accuracy).** In Fig. A.1a, the best-so-far OOD accuracy rises sharply in the early phase and then shows a saturation trend, indicating that SPARK can convert a small candidate budget into substantial performance gains, while later iterations provide more incremental improvements. The thick mean curve together with the min–max band suggests this pattern is consistent across seeds rather than being driven by a single favorable run. We attribute the strong early gains to SPARK's structured (scope-local) updates: by committing each iteration to an explicit scope via ASR and applying RC-guided, SAR-executed edits within a constrained feasible set, the search produces composable improvements that accumulate across rounds instead of being washed out by unconstrained rewrites.

**Cost control (MACs of the incumbent best model).** Fig. A.1b tracks the MACs of the incumbent best-so-far model throughout the same 100-attempt trajectory. Although MACs may change when the incumbent is replaced, the curves do not exhibit a systematic upward trend and remain within a stable range across seeds. This indicates that the accuracy gains in Fig. A.1a are not primarily explained by progressively increasing compute, but rather by discovering more effective structures within a comparable compute envelope—consistent with SPARK's goal

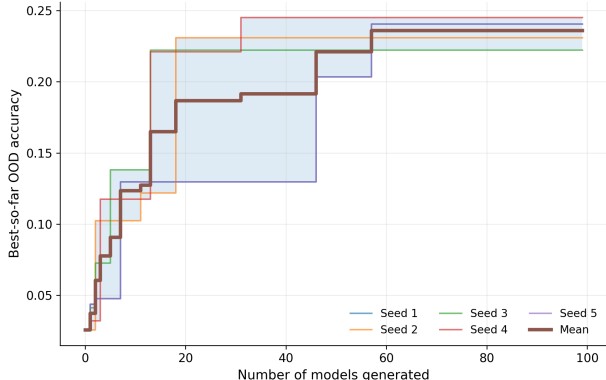

*(a)* Search progress measured by best-so-far OOD accuracy.

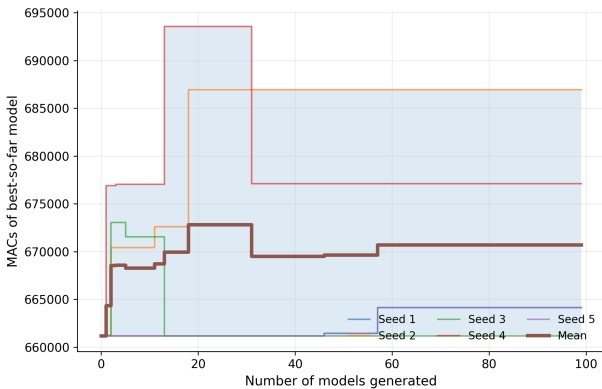

*(b)* Compute footprint of the best-so-far model measured by MACs.

*Figure A.1.* Search dynamics on CLRS over 100 generated candidates (5 seeds). Lines show best-so-far trajectories per seed; the thick line is the mean and the shaded band indicates the min–max range. (a) Best-so-far OOD accuracy rises quickly then saturates. (b) Best-so-far MACs remain stable, suggesting gains come from improved structures rather than increasing compute.

of structure-guided refinement rather than cost-increasing scaling.

**Why both plots.** We report both progress and cost because reliable multi-round LLM-driven NAS requires improvements to be (i) attributable and composable across iterations (reflected by best-so-far accuracy gains) while also being (ii) cost-bounded (reflected by stable MACs). Together, Fig. A.1a and Fig. A.1b provide complementary evidence that SPARK achieves effective, budget-efficient progress without relying on an expanding computational footprint.

## A.3. Metric Definitions

We report search-cost and benchmark statistics using the following metrics. **Cand.** denotes the number of proposed candidates generated by the search procedure. **LLM calls** denotes the total number of LLM invocations, including calls made during routing, directive generation, and code

*Table A.1.* Per-task OOD accuracy (%) on 30 CLRS tasks.

| Task | CLRS | EP | Ours | Task | CLRS | EP | Ours |
|------|------|------|------|------|------|------|------|
| DFS | 47.8 | 68.1 | **83.7** | MCO | **91.7** | 90.8 | 87.3 |
| AP | 88.3 | 93.5 | **97.9** | MPrim | 86.4 | **88.7** | **88.7** |
| QS | 0.5 | 0.8 | **10.7** | SI | 97.6 | **98.2** | 97.1 |
| SCC | 43.4 | 41.9 | **77.3** | BS | 77.6 | 78.0 | **87.6** |
| ASel | 95.2 | 95.1 | **98.0** | Br | 94.0 | 97.6 | **98.2** |
| FMS | 76.4 | 75.4 | **85.6** | Bub | 67.7 | 88.9 | **98.1** |
| Dij | 96.1 | 97.3 | **99.2** | GS | 93.6 | 93.8 | **96.4** |
| LCS | 80.5 | 85.8 | **87.6** | HS | 31.0 | 69.9 | **75.6** |
| Min | 97.8 | 98.4 | **99.1** | Ins | 78.1 | 89.5 | **97.5** |
| Topo | 87.3 | 88.1 | **99.7** | JM | 91.0 | 90.4 | **94.9** |
| BFS | 99.7 | **100.0** | 70.6 | KMP | **19.5** | 16.3 | 14.9 |
| FW | 48.5 | **61.4** | 32.9 | MKr | 89.8 | **91.5** | 86.2 |
| TSch | 87.3 | **88.2** | 87.4 | NSM | 78.7 | **79.8** | 77.3 |
| BFord | 97.4 | **97.5** | 96.0 | OBST | 73.8 | 78.7 | **96.1** |
| DSP | **98.2** | 98.0 | 97.8 | QSort | 64.6 | 85.2 | **97.8** |

**Abbreviations:** AP=Articulation Points, QS=Quickselect, SCC=Strongly Connected Components, ASel=Activity Selector, FMS=Find Maximum Subarray (Kadane), Dij=Dijkstra, LCS=LCS Length, Min=Minimum, Topo=Topological Sort, BFS=Breadth-First Search, FW=Floyd–Warshall, TSch=Task Scheduling, BFord=Bellman–Ford, DSP=DAG Shortest Paths, MCO=Matrix Chain Order, MPrim=MST Prim, SI=Segments Intersect, BS=Binary Search, Br=Bridges, Bub=Bubble Sort, GS=Graham Scan, HS=Heapsort, Ins=Insertion Sort, JM=Jarvis March, KMP=KMP Matcher, MKr=MST Kruskal, NSM=Naive String Matcher, OBST=Optimal BST, QSort=Quicksort.

editing. For SPARK, one proposal may involve multiple role-specific calls because each evolution step is decomposed into scope routing, directive construction, and scoped refinement. **GPU-hrs** and **Wall-h** measure the search-time cost. **Train(s)** denotes the average training time per evaluated candidate. **MACs** measures the model-level compute of the final selected architecture. Thus, MACs should be interpreted as architecture-level compute, whereas GPU-hours, wall-clock time, and LLM calls reflect the cost of the search process.

### A.4. Scope-Control Baselines and Step-Level Example

We further examine whether SPARK's gains come merely from adding explicit region tags, or from the complete interaction among scope selection, directive generation, and scoped refinement. This diagnostic ablation uses the same editor setting as the Qwen-Plus block in Table 3; therefore, the Full SPARK row corresponds to the 80.50 DFS OOD accuracy reported there. The compared variants progressively add or remove components: Global Rewrite performs unconstrained rewriting; Random-Scope SAR uses the region/tag mechanism but chooses the editable scope randomly; Fixed-OP SAR and Fixed-ACT SAR always edit a fixed scope; Random RC+SAR keeps directive generation and scoped refinement but removes learned scope routing; ASR+SAR keeps scope routing and scoped refinement but removes the RC directive; Full SPARK uses the complete ASR+RC+SAR pipeline.

Table A.2 shows that region tags alone are not sufficient.

Random-Scope SAR improves over global rewriting but remains far below Full SPARK (50.27 vs. 80.50), indicating that simply constraining edits to a tagged region does not provide reliable search guidance. The fixed-scope variants are also weak: Fixed-OP SAR stays close to the CLRS reference, while Fixed-ACT SAR degrades performance, suggesting that a static choice of edit region cannot adapt to the current search state. Adding RC+SAR with a random scope improves accuracy to 57.84, and using ASR+SAR without RC further improves it to 65.93, showing that both scope selection and directive-conditioned refinement contribute complementary gains. Full SPARK achieves the best accuracy with comparable MACs, supporting that the improvement comes from the complete factor-conditioned pipeline rather than from region tags alone.

**Illustrative SPARK step linked to the evolved program.** Table A.3 illustrates a representative SPARK step that contributes to the evolved behavior later analyzed in Listing 5. The example focuses on the ACTION-side refinement in the triplet-message computation: the search context suggests that the current program needs better topology-aware masking and more stable aggregation, while the operator definitions should remain fixed for this step. Listing 5 then shows the resulting evolved code state in the trajectory, where the triplet-message computation incorporates adjacency-aware masking and multi-statistic aggregation.

*Table A.2.* Scope-control ablations on CLRS-DFS. Random-Scope SAR serves as a region/tag-only baseline; Fixed-OP SAR and Fixed-ACT SAR test fixed-scope editing; Random RC+SAR removes learned scope routing; ASR+SAR removes the RC directive; Full SPARK uses the complete ASR+RC+SAR pipeline. MACs is model compute, Acc. is DFS OOD accuracy, and $\Delta$ is the gain in percentage points over the CLRS reference.

| Variant | Controlled component | MACs↓ | Acc.↑ | $\Delta$↑ |
|---|---|---|---|---|
| CLRS reference | No search/edit | 661,190 | 46.80 | +0.00 |
| Global Rewrite | Unconstrained rewrite | 661,190 | 39.43 | -7.37 |
| Random-Scope SAR | Region/tag-only, random scope | 661,422 | 50.27 | +3.47 |
| Fixed-OP SAR | Fixed OPERATOR scope | 661,190 | 46.31 | -0.49 |
| Fixed-ACT SAR | Fixed ACTION scope | 655,039 | 41.67 | -5.13 |
| Random RC+SAR | RC+SAR with random scope | 655,014 | 57.84 | +11.04 |
| ASR+SAR | ASR-guided scope, no RC | 661,390 | 65.93 | +19.13 |
| Full SPARK | ASR+RC+SAR | 665,308 | **80.50** | **+33.70** |

*Table A.3.* Illustrative one-step execution of SPARK, linked to the program evolution analyzed in Listing 5. The example shows how ASR, RC, and SAR decompose an architecture update into scope selection, directive generation, and factor-local code editing.

| Stage | Content |
|---|---|
| Search context | The parent program already has a stable triplet-message pathway, but recent candidates show limited progress. The likely bottleneck is not the existence of projection operators, but how triplet messages are composed, masked, and aggregated in the forward computation. |
| ASR: scope routing | Select ACTION. The next edit should modify the message-computation behavior in `_compute_tri_msgs`, while keeping the non-selected OPERATOR region unchanged for this step. |
| RC: refinement directive | Produce a scope-local directive: preserve public interfaces and tensor contracts; refine the ACTION region by using graph topology to suppress invalid non-edge messages; combine complementary aggregation statistics to improve stability; keep the output shape compatible with the CLRS processor. |
| SAR: scoped edit | Generate an ACTION-local patch that updates the triplet-message computation. The patch applies adjacency-aware masking, combines max- and mean-style aggregation, and keeps the message tensor aligned with the expected node-pair shape. |
| Feasibility check | Verify that region tags are preserved, the edit is restricted to the selected scope, required symbols and signatures remain valid, and a forward-only check passes tensor-shape and masking constraints before full evaluation. |
| Connection to Listing 5 | Listing 5 shows the evolved program state containing this type of topology-aware masking and multi-statistic triplet aggregation. The table explains the corresponding SPARK decision process, while the listing shows the concrete code-level realization in the evolution trace. |

## A.5. Additional Benchmark Protocol

To evaluate whether SPARK is specific to CLRS, we further conduct experiments on five non-CLRS benchmarks spanning different problem types: MNIST-1D (Greydanus & Kobak, 2020) for lightweight classification, 20News (Mitchell, 1999) for text classification, CartPole-v1 (Gavenski et al., 2024) for reinforcement learning, Mice Protein Expression (Higuera et al., 2015) for protein-expression classification, and Breast Cancer Wisconsin (Wolberg et al., 1993) for medical diagnosis. Across these benchmarks, we use the same high-level LLM-driven search interface and compare against OpenEvolve, Evo-Prompting, FunSearch, and EoH when applicable. We report task performance together with search-cost statistics, including the number of proposed candidates, wall-clock time, per-candidate training time, and model-level MACs.

These benchmarks are not intended to replace the full CLRS evaluation, but to test whether the proposed factor-conditioned editing mechanism can be instantiated beyond the original algorithmic-reasoning setting.

The main text summarizes the additional benchmark results in Table 2. Here, Table A.4 reports the MNIST-1D results across four backbone architectures, and Table A.5 provides detailed search-cost and performance statistics on 20News, CartPole-v1, Mice Protein Expression, and Breast Cancer Wisconsin.

**Discussion.** The additional benchmark results in Table 2, together with the detailed statistics in Tables A.4 and A.5, show that SPARK can be instantiated beyond CLRS-style algorithmic reasoning programs. On MNIST-1D, SPARK consistently improves all four backbone architectures and achieves the highest average accuracy. On 20News and

*Table A.4.* Additional results on MNIST-1D across four backbone architectures. Each cell reports accuracy/loss. Avg. Acc. is averaged over the four backbones.

| Method | LR | MLP | CNN | GRU | Avg. Acc. |
|---|---|---|---|---|---|
| Base | 37.0 / 1.67 | 64.0 / 3.01 | 93.2 / 0.47 | 88.9 / 0.66 | 71.05 |
| EoH | 93.0 / 0.22 | 64.0 / 3.06 | 93.0 / 0.49 | 67.1 / 0.70 | 79.90 |
| OpenEvolve | 61.0 / 1.31 | 65.8 / 1.82 | 94.7 / 0.27 | 59.5 / 3.09 | 70.25 |
| FunSearch | 29.5 / 1.70 | 64.8 / 3.06 | 95.0 / 0.27 | 89.7 / 0.67 | 69.75 |
| EvoPrompting | 65.7 / 2.15 | 71.2 / 1.49 | 95.6 / 0.42 | 61.1 / 1.39 | 73.40 |
| Ours | **98.4 / 0.13** | **88.36 / 0.99** | **98.1 / 0.08** | **99.1 / 0.08** | **95.99** |

*Table A.5.* Detailed results on additional benchmarks beyond CLRS: 20News (Mitchell, 1999), Mice Protein Expression (Higuera et al., 2015), Breast Cancer Wisconsin (Wolberg et al., 1993), and CartPole-v1 (Gavenski et al., 2024). OE, EP, and FS denote OpenEvolve, EvoPrompting, and FunSearch. Cand. denotes proposed candidates; Wall-h is search wall-clock time; Train(s) is average per-candidate training time; MACs(K) is model-level compute.

| Benchmark | Metric | Base | OE | EP | EoH | FS | Ours |
|---|---|---|---|---|---|---|---|
| *20News*
Text cls. | Cand. | – | 100 | 100 | 400 | 400 | 100 |
| | Wall-h | – | 0.29 | 0.28 | 1.15 | 1.23 | 0.51 |
| | Train(s) | 6.60 | 7.84 | 6.70 | 6.83 | 7.36 | 6.86 |
| | MACs(K) | 68.24 | 75.61 | 68.50 | 68.50 | 75.60 | 68.76 |
| | Acc. (%) ↑ | 50.50 | 57.10 | 58.01 | 58.05 | 57.47 | **59.13** |
| | Loss ↓ | 1.76 | 1.55 | **1.47** | **1.47** | 1.50 | 1.59 |
| *Mice Protein Expression*
Protein cls. | Cand. | – | 100 | 100 | 400 | 400 | 100 |
| | Wall-h | – | 0.10 | 0.21 | 0.69 | 0.67 | 0.30 |
| | Train(s) | 3.02 | 4.83 | 4.64 | 2.55 | 3.02 | 4.75 |
| | MACs(K) | 1.674 | 1.646 | 5.248 | 2.624 | 1.674 | 1.886 |
| | Acc. (%) ↑ | 58.92 | **61.78** | 61.62 | 58.25 | 58.92 | **61.78** |
| | Loss ↓ | 1.06 | 1.08 | 1.05 | 1.05 | 1.06 | **1.03** |
| *Breast Cancer Wisconsin*
Medical diag. | Cand. | – | 100 | 100 | 400 | 400 | 100 |
| | Wall-h | – | 0.13 | 0.18 | 0.56 | 0.51 | 0.24 |
| | Train(s) | 1.71 | 1.92 | 1.74 | 1.68 | 1.71 | 1.83 |
| | MACs(K) | 3.202 | 3.202 | 3.202 | 7.586 | 3.202 | 3.390 |
| | Acc. (%) ↑ | 95.13 | 96.90 | 96.90 | 96.46 | 95.13 | **97.04** |
| | Loss ↓ | **0.13** | 0.97 | 0.97 | 0.14 | **0.13** | 0.16 |
| *CartPole-v1*
RL | Cand. | – | 100 | 100 | 400 | 400 | 100 |
| | Wall-h | – | 0.73 | 0.47 | 3.54 | 3.94 | 2.13 |
| | Train(s) | 14.40 | 19.01 | 19.48 | 28.42 | 62.37 | 67.18 |
| | MACs(K) | 0.226 | 2.466 | 0.450 | 0.146 | 4.610 | 2.690 |
| | Test Ret. ↑ | 121.60 | 377.55 | 488.65 | 31.20 | 276.28 | **489.00** |
| | Train Ret. ↑ | 17.39 | 48.01 | 54.26 | 139.58 | 119.75 | **175.15** |

Breast Cancer Wisconsin, SPARK yields modest but consistent improvements over the compared baselines. On CartPole-v1, SPARK obtains the highest test return, improving over the base policy and slightly outperforming the strongest baseline under the same 100-candidate budget. On Mice Protein Expression, SPARK matches the best accuracy and achieves the lowest loss among the compared methods. Overall, these results suggest that factor-conditioned editing is applicable across classification, text classification, reinforcement learning, protein-expression classification, and medical diagnosis settings, while its additional search-time overhead should be interpreted together with the final performance gains.

### A.6. Program Evolution Analysis Across Iterations

This section analyzes how the PGN triplet module evolves across iterations. We treat `_maybe_lazy_init` as the Operator (parameterization/structure definition) and `_compute_tri_msgs` as the Action (triplet message computation). Iteration 0 is the base program we aim to evolve (Listing 1); later iterations introduce increasingly expressive, stable, and structure-aware computation.

**Iteration 0 (Base): additive triplet construction with max aggregation.** As shown in Listing 1, the base Operator instantiates linear projections for node features (`z`), edge features (`edge_fts`), and graph-level features (`graph_fts`). The base Action constructs a triplet tensor via broadcast-

ing (`unsqueeze`) and combines components primarily through additive composition followed by a max-style reduction. This design is simple and stable, providing a strong baseline with clear gradient paths and a reliable anchor for subsequent evolution.

**Iteration 1: multi-head parameterization and explicit interaction mixing.** Compared with the base, Iteration 1 (Listing 2) evolves the Operator to a multi-head design (e.g., per-head projections and head-wise composition), enabling multiple latent "views" of triplet reasoning. In the Action, the computation moves beyond purely additive composition by introducing explicit interaction mixing terms (e.g., node–node, node–edge, and edge–edge interactions) before head fusion. This change increases compositional expressivity: multiple heads can capture diverse relational patterns and reduce the bottleneck of a single projection path.

**Iteration 2: residual augmentation for more stable learning.** Iteration 2 (Listing 3) keeps the base-style aggregation backbone but introduces a residual pathway in the Action, injecting an additional direct mapping (from a selected intermediate component) into the output. Benefit: improved optimization stability—when triplet aggregation is uncertain early in training, the residual provides a reliable shortcut signal, mitigating under-training of deep interaction paths.

**Iteration 7: gated residual injection to suppress noisy edges.** Building on Iteration 2, Iteration 7 (Listing 4) adds a gating mechanism on the residual branch, so the residual contribution becomes evidence-dependent rather than always-on. Benefit: better robustness—noisy or low-confidence edges are less likely to dominate the message update, while informative edges retain strong influence.

**Iteration 46: structure-aware masking and multi-statistic aggregation.** Iteration 46 (Listing 5) introduces a structural prior into the Action by incorporating an adjacency mask (e.g., `adj_mat`) to constrain where messages are valid, effectively preventing propagation along non-existent edges. It also enriches the aggregation by combining complementary statistics (e.g., max- and mean-style summaries) followed by a learned mixer. Benefit: (i) improved structural consistency—messages respect the graph topology; (ii) more robust summarization—max captures salient evidence while mean stabilizes noisy activations.

**Iteration 57: context-conditioned channel gating for node/edge/global streams.** Iteration 57 (Listing 6) evolves the Operator into a more adaptive form by adding fine-grained gates over the node, edge, and global feature streams, making projections condition-dependent. The Action correspondingly applies these gates to modulate which channels/branches are emphasized under different inputs, and further conditions residual control on richer context. Benefit: strong adaptivity and generalization—dynamic channel selection acts as a soft routing mechanism, allowing the module to emphasize the most relevant relational cues under varying structural and contextual conditions.

**Overall trend.** Starting from the base (Iteration 0), the evolution follows a clear trajectory: (1) expressivity via multi-head and interaction mixing (Iteration 1), (2) stability via residual and gated residual designs (Iterations 2 and 7), and (3) validity & adaptivity via topology-aware masking and context-conditioned gating (Iterations 46 and 57). These changes progressively strengthen the Operator (better parameterization and controllability) and the Action (more robust and structure-consistent triplet message computation), aligning the evolved program with the needs of reliable graph reasoning.

*Listing 1.* Iteration 0: Operator _maybe_lazy_init and Action _compute_tri_msgs

```
1  def _maybe_lazy_init(self, z: _Array, edge_fts: _Array, graph_fts: _Array):
2      in_z = z.shape[-1]
3      in_e = edge_fts.shape[-1]
4      in_g = graph_fts.shape[-1]
5      if self.use_triplets:
6
7          self.t_1 = nn.Linear(in_z, self.nb_triplet_fts)
8          self.t_2 = nn.Linear(in_z, self.nb_triplet_fts)
9          self.t_3 = nn.Linear(in_z, self.nb_triplet_fts)
10         self.t_e_1 = nn.Linear(in_e, self.nb_triplet_fts)
11         self.t_e_2 = nn.Linear(in_e, self.nb_triplet_fts)
12         self.t_e_3 = nn.Linear(in_e, self.nb_triplet_fts)
13         self.t_g = nn.Linear(in_g, self.nb_triplet_fts)
14         self.o3 = nn.Linear(self.nb_triplet_fts, self.out_size)
15
16  def _compute_tri_msgs(
17      self,
18      z: _Array,
19      edge_fts: _Array,
20      graph_fts: _Array,
21  ) -> Optional[_Array]:
22
23
24
25      if not self.use_triplets:
26          return None
27
28
29      tri_1 = self.t_1(z)
30      tri_2 = self.t_2(z)
31      tri_3 = self.t_3(z)
32      tri_e_1 = self.t_e_1(edge_fts)
33      tri_e_2 = self.t_e_2(edge_fts)
34      tri_e_3 = self.t_e_3(edge_fts)
35      tri_g = self.t_g(graph_fts)
36
37      triplets = (
38          tri_1.unsqueeze(2).unsqueeze(3) +
39          tri_2.unsqueeze(1).unsqueeze(3) +
40          tri_3.unsqueeze(1).unsqueeze(2) +
41          tri_e_1.unsqueeze(3) +
42          tri_e_2.unsqueeze(2) +
43          tri_e_3.unsqueeze(1) +
44          tri_g.unsqueeze(1).unsqueeze(2).unsqueeze(3)
45      )
46      tri_msgs = self.o3(torch.amax(triplets, dim=1))
47      if self.activation is not None:
48          tri_msgs = self.activation(tri_msgs)
49
50
51      return tri_msgs
```

*Listing 2.* Iteration 1: Operator _maybe_lazy_init and Action _compute_tri_msgs

```
1  def _maybe_lazy_init(self, z: _Array, edge_fts: _Array, graph_fts: _Array):
2      in_z = z.shape[-1]
3      in_e = edge_fts.shape[-1]
4      in_g = graph_fts.shape[-1]
5      if self.use_triplets:
6
7          head_dim = self.nb_triplet_fts // 4
8          self.heads = nn.ModuleList([nn.ModuleDict({
9              't_1': nn.Linear(in_z, head_dim),
```

```
10          't_2': nn.Linear(in_z, head_dim),
11          't_3': nn.Linear(in_z, head_dim),
12          't_e_1': nn.Linear(in_e, head_dim),
13          't_e_2': nn.Linear(in_e, head_dim),
14          't_e_3': nn.Linear(in_e, head_dim),
15          't_g': nn.Linear(in_g, head_dim)
16      }) for _ in range(4)])
17      self.o3 = nn.Linear(4 * head_dim, self.out_size)
18
19  def _compute_tri_msgs(
20      self,
21      z: _Array,
22      edge_fts: _Array,
23      graph_fts: _Array,
24  ) -> Optional[_Array]:
25
26      if not self.use_triplets:
27          return None
28
29
30      head_outputs = []
31      for head in self.heads:
32          tri_1 = head['t_1'](z)
33          tri_2 = head['t_2'](z)
34          tri_3 = head['t_3'](z)
35          tri_e_1 = head['t_e_1'](edge_fts)
36          tri_e_2 = head['t_e_2'](edge_fts)
37          tri_e_3 = head['t_e_3'](edge_fts)
38          tri_g = head['t_g'](graph_fts)
39
40
41          mix_i = tri_1.unsqueeze(2) * tri_2.unsqueeze(1)
42          mix_j = tri_3.unsqueeze(1) * tri_e_2
43          mix_k = tri_e_1 * tri_e_3
44
45
46          head_tri = mix_i + mix_j + mix_k + tri_g.unsqueeze(1).unsqueeze(2)
47          head_outputs.append(head_tri)
48
49
50      triplets = torch.cat(head_outputs, dim=-1)
51      tri_msgs = self.o3(triplets)
52
53      if self.activation is not None:
54          tri_msgs = self.activation(tri_msgs)
55
56
57      return tri_msgs
```

*Listing 3.* Iteration 2: Operator _maybe_lazy_init and Action _compute_tri_msgs

```
1  def _maybe_lazy_init(self, z: _Array, edge_fts: _Array, graph_fts: _Array):
2      in_z = z.shape[-1]
3      in_e = edge_fts.shape[-1]
4      in_g = graph_fts.shape[-1]
5      if self.use_triplets:
6
7          half1 = self.nb_triplet_fts // 2
8          half2 = self.nb_triplet_fts - half1
9
10
11          self.t_1_head1 = nn.Linear(in_z, half1)
12          self.t_1_head2 = nn.Linear(in_z, half2)
13          self.t_2_head1 = nn.Linear(in_z, half1)
```

```
14      self.t_2_head2 = nn.Linear(in_z, half2)
15      self.t_3_head1 = nn.Linear(in_z, half1)
16      self.t_3_head2 = nn.Linear(in_z, half2)
17
18
19      self.t_e_1_head1 = nn.Linear(in_e, half1)
20      self.t_e_1_head2 = nn.Linear(in_e, half2)
21      self.t_e_2_head1 = nn.Linear(in_e, half1)
22      self.t_e_2_head2 = nn.Linear(in_e, half2)
23      self.t_e_3_head1 = nn.Linear(in_e, half1)
24      self.t_e_3_head2 = nn.Linear(in_e, half2)
25
26
27      self.t_g_head1 = nn.Linear(in_g, half1)
28      self.t_g_head2 = nn.Linear(in_g, half2)
29
30
31      self.o3 = nn.Linear(self.nb_triplet_fts, self.out_size)
32
33  def _compute_tri_msgs(
34      self,
35      z: _Array,
36      edge_fts: _Array,
37      graph_fts: _Array,
38  ) -> Optional[_Array]:
39
40
41
42    if not self.use_triplets:
43      return None
44
45
46
47    tri_1 = torch.cat([self.t_1_head1(z), self.t_1_head2(z)], dim=-1)
48    tri_2 = torch.cat([self.t_2_head1(z), self.t_2_head2(z)], dim=-1)
49    tri_3 = torch.cat([self.t_3_head1(z), self.t_3_head2(z)], dim=-1)
50    tri_e_1 = torch.cat([self.t_e_1_head1(edge_fts), self.t_e_1_head2(edge_fts)], dim
        ↪ =-1)
51    tri_e_2 = torch.cat([self.t_e_2_head1(edge_fts), self.t_e_2_head2(edge_fts)], dim
        ↪ =-1)
52    tri_e_3 = torch.cat([self.t_e_3_head1(edge_fts), self.t_e_3_head2(edge_fts)], dim
        ↪ =-1)
53    tri_g = torch.cat([self.t_g_head1(graph_fts), self.t_g_head2(graph_fts)], dim=-1)
54
55    triplets = (
56        tri_1.unsqueeze(2).unsqueeze(3) +
57        tri_2.unsqueeze(1).unsqueeze(3) +
58        tri_3.unsqueeze(1).unsqueeze(2) +
59        tri_e_1.unsqueeze(3) +
60        tri_e_2.unsqueeze(2) +
61        tri_e_3.unsqueeze(1) +
62        tri_g.unsqueeze(1).unsqueeze(2).unsqueeze(3)
63    )
64
65
66    tri_msgs_main = self.o3(torch.amax(triplets, dim=1))
67
68
69    residual = self.o3(F.relu(tri_e_1))
70    tri_msgs = tri_msgs_main + residual
71
72    if self.activation is not None:
73      tri_msgs = self.activation(tri_msgs)
74
75
```

```
76        return tri_msgs
```

*Listing 4.* Iteration 7: Operator _maybe_lazy_init and Action _compute_tri_msgs

```
1  def _maybe_lazy_init(self, z: _Array, edge_fts: _Array, graph_fts: _Array):
2      in_z = z.shape[-1]
3      in_e = edge_fts.shape[-1]
4      in_g = graph_fts.shape[-1]
5      if self.use_triplets:
6
7        half1 = self.nb_triplet_fts // 2
8        half2 = self.nb_triplet_fts - half1
9
10
11       self.t_1_head1 = nn.Linear(in_z, half1)
12       self.t_1_head2 = nn.Linear(in_z, half2)
13       self.t_2_head1 = nn.Linear(in_z, half1)
14       self.t_2_head2 = nn.Linear(in_z, half2)
15       self.t_3_head1 = nn.Linear(in_z, half1)
16       self.t_3_head2 = nn.Linear(in_z, half2)
17
18
19       self.t_e_1_head1 = nn.Linear(in_e, half1)
20       self.t_e_1_head2 = nn.Linear(in_e, half2)
21       self.t_e_2_head1 = nn.Linear(in_e, half1)
22       self.t_e_2_head2 = nn.Linear(in_e, half2)
23       self.t_e_3_head1 = nn.Linear(in_e, half1)
24       self.t_e_3_head2 = nn.Linear(in_e, half2)
25
26
27       self.t_g_head1 = nn.Linear(in_g, half1)
28       self.t_g_head2 = nn.Linear(in_g, half2)
29
30
31       self.o3 = nn.Linear(self.nb_triplet_fts, self.out_size)
32
33    def _compute_tri_msgs(
34        self,
35        z: _Array,
36        edge_fts: _Array,
37        graph_fts: _Array,
38    ) -> Optional[_Array]:
39
40
41
42      if not self.use_triplets:
43        return None
44
45
46
47      tri_1 = torch.cat([self.t_1_head1(z), self.t_1_head2(z)], dim=-1)
48      tri_2 = torch.cat([self.t_2_head1(z), self.t_2_head2(z)], dim=-1)
49      tri_3 = torch.cat([self.t_3_head1(z), self.t_3_head2(z)], dim=-1)
50      tri_e_1 = torch.cat([self.t_e_1_head1(edge_fts), self.t_e_1_head2(edge_fts)], dim
         ↪ =-1)
51      tri_e_2 = torch.cat([self.t_e_2_head1(edge_fts), self.t_e_2_head2(edge_fts)], dim
         ↪ =-1)
52      tri_e_3 = torch.cat([self.t_e_3_head1(edge_fts), self.t_e_3_head2(edge_fts)], dim
         ↪ =-1)
53      tri_g = torch.cat([self.t_g_head1(graph_fts), self.t_g_head2(graph_fts)], dim=-1)
54
55      triplets = (
56          tri_1.unsqueeze(2).unsqueeze(3) +
57          tri_2.unsqueeze(1).unsqueeze(3) +
```

```
58          tri_3.unsqueeze(1).unsqueeze(2) +
59          tri_e_1.unsqueeze(3) +
60          tri_e_2.unsqueeze(2) +
61          tri_e_3.unsqueeze(1) +
62          tri_g.unsqueeze(1).unsqueeze(2).unsqueeze(3)
63      )
64
65
66      tri_msgs_main = self.o3(torch.amax(triplets, dim=1))
67
68
69      residual = self.o3(F.relu(tri_e_1))
70      gate = torch.sigmoid(edge_fts.mean(dim=-1, keepdim=True))
71      residual = residual * gate
72
73      tri_msgs = tri_msgs_main + residual
74
75      if self.activation is not None:
76        tri_msgs = self.activation(tri_msgs)
77
78
79      return tri_msgs
```

*Listing 5.* Iteration 46: Operator _maybe_lazy_init and Action _compute_tri_msgs

```
1   def _maybe_lazy_init(self, z: _Array, edge_fts: _Array, graph_fts: _Array):
2       in_z = z.shape[-1]
3       in_e = edge_fts.shape[-1]
4       in_g = graph_fts.shape[-1]
5       if self.use_triplets:
6         quarter = self.nb_triplet_fts // 4
7         quarters = [quarter] * 4
8         quarters[3] = self.nb_triplet_fts - 3 * quarter
9
10        self.t_1_head1 = nn.Linear(in_z, quarters[0])
11        self.t_1_head2 = nn.Linear(in_z, quarters[1])
12        self.t_1_head3 = nn.Linear(in_z, quarters[2])
13        self.t_1_head4 = nn.Linear(in_z, quarters[3])
14        self.t_2_head1 = nn.Linear(in_z, quarters[0])
15        self.t_2_head2 = nn.Linear(in_z, quarters[1])
16        self.t_2_head3 = nn.Linear(in_z, quarters[2])
17        self.t_2_head4 = nn.Linear(in_z, quarters[3])
18        self.t_3_head1 = nn.Linear(in_z, quarters[0])
19        self.t_3_head2 = nn.Linear(in_z, quarters[1])
20        self.t_3_head3 = nn.Linear(in_z, quarters[2])
21        self.t_3_head4 = nn.Linear(in_z, quarters[3])
22
23
24        self.t_e_1_head1 = nn.Linear(in_e, quarters[0])
25        self.t_e_1_head2 = nn.Linear(in_e, quarters[1])
26        self.t_e_1_head3 = nn.Linear(in_e, quarters[2])
27        self.t_e_1_head4 = nn.Linear(in_e, quarters[3])
28        self.t_e_2_head1 = nn.Linear(in_e, quarters[0])
29        self.t_e_2_head2 = nn.Linear(in_e, quarters[1])
30        self.t_e_2_head3 = nn.Linear(in_e, quarters[2])
31        self.t_e_2_head4 = nn.Linear(in_e, quarters[3])
32        self.t_e_3_head1 = nn.Linear(in_e, quarters[0])
33        self.t_e_3_head2 = nn.Linear(in_e, quarters[1])
34        self.t_e_3_head3 = nn.Linear(in_e, quarters[2])
35        self.t_e_3_head4 = nn.Linear(in_e, quarters[3])
36
37
38        self.t_g_head1 = nn.Linear(in_g, quarters[0])
39        self.t_g_head2 = nn.Linear(in_g, quarters[1])
```

```
40       self.t_g_head3 = nn.Linear(in_g, quarters[2])
41       self.t_g_head4 = nn.Linear(in_g, quarters[3])
42
43
44       self.o3 = nn.Linear(self.nb_triplet_fts, self.out_size)
45
46
47       self.aggr_mixer = nn.Linear(2 * self.nb_triplet_fts, self.nb_triplet_fts)
48
49       self.res_gate = nn.Linear(in_e, 1)
50
51   def _compute_tri_msgs(
52       self,
53       z: _Array,
54       edge_fts: _Array,
55       graph_fts: _Array,
56       adj_mat: _Array,
57   ) -> Optional[_Array]:
58
59     if not self.use_triplets:
60       return None
61
62
63
64     tri_1 = torch.cat([
65         self.t_1_head1(z), self.t_1_head2(z),
66         self.t_1_head3(z), self.t_1_head4(z)
67     ], dim=-1)
68     tri_2 = torch.cat([
69         self.t_2_head1(z), self.t_2_head2(z),
70         self.t_2_head3(z), self.t_2_head4(z)
71     ], dim=-1)
72     tri_3 = torch.cat([
73         self.t_3_head1(z), self.t_3_head2(z),
74         self.t_3_head3(z), self.t_3_head4(z)
75     ], dim=-1)
76     tri_e_1 = torch.cat([
77         self.t_e_1_head1(edge_fts), self.t_e_1_head2(edge_fts),
78         self.t_e_1_head3(edge_fts), self.t_e_1_head4(edge_fts)
79     ], dim=-1)
80     tri_e_2 = torch.cat([
81         self.t_e_2_head1(edge_fts), self.t_e_2_head2(edge_fts),
82         self.t_e_2_head3(edge_fts), self.t_e_2_head4(edge_fts)
83     ], dim=-1)
84     tri_e_3 = torch.cat([
85         self.t_e_3_head1(edge_fts), self.t_e_3_head2(edge_fts),
86         self.t_e_3_head3(edge_fts), self.t_e_3_head4(edge_fts)
87     ], dim=-1)
88     tri_g = torch.cat([
89         self.t_g_head1(graph_fts), self.t_g_head2(graph_fts),
90         self.t_g_head3(graph_fts), self.t_g_head4(graph_fts)
91     ], dim=-1)
92
93
94     triplets = (
95         tri_1.unsqueeze(2).unsqueeze(3) +
96         tri_2.unsqueeze(1).unsqueeze(3) +
97         tri_3.unsqueeze(1).unsqueeze(2) +
98         tri_e_1.unsqueeze(3) +
99         tri_e_2.unsqueeze(2) +
100        tri_e_3.unsqueeze(1) +
101        tri_g.unsqueeze(1).unsqueeze(2).unsqueeze(3)
102    )
103
104
```

```
105    max_aggr = torch.amax(triplets, dim=3)
106    mean_aggr = torch.mean(triplets, dim=3)
107    combined = torch.cat([max_aggr, mean_aggr], dim=-1)
108    mixed_aggr = self.aggr_mixer(combined)
109    tri_msgs_main = self.o3(mixed_aggr)
110
111
112    gate = torch.sigmoid(self.res_gate(edge_fts)) * adj_mat.unsqueeze(-1)
113    residual = self.o3(F.relu(tri_e_1)) * gate
114    tri_msgs = tri_msgs_main + residual
115
116
117    tri_msgs = tri_msgs * adj_mat.unsqueeze(-1)
118
119    if self.activation is not None:
120      tri_msgs = self.activation(tri_msgs)
121
122
123    return tri_msgs
```

*Listing 6.* Iteration 57: Operator \_maybe\_lazy\_init and Action \_compute\_tri\_msgs

```
1  def _maybe_lazy_init(self, z: _Array, edge_fts: _Array, graph_fts: _Array):
2      in_z = z.shape[-1]
3      in_e = edge_fts.shape[-1]
4      in_g = graph_fts.shape[-1]
5      if self.use_triplets:
6
7        half1 = self.nb_triplet_fts // 2
8        half2 = self.nb_triplet_fts - half1
9
10
11        self.t_1_head1 = nn.Linear(in_z, half1)
12        self.t_1_head2 = nn.Linear(in_z, half2)
13        self.t_2_head1 = nn.Linear(in_z, half1)
14        self.t_2_head2 = nn.Linear(in_z, half2)
15        self.t_3_head1 = nn.Linear(in_z, half1)
16        self.t_3_head2 = nn.Linear(in_z, half2)
17
18
19        self.t_e_1_head1 = nn.Linear(in_e, half1)
20        self.t_e_1_head2 = nn.Linear(in_e, half2)
21        self.t_e_2_head1 = nn.Linear(in_e, half1)
22        self.t_e_2_head2 = nn.Linear(in_e, half2)
23        self.t_e_3_head1 = nn.Linear(in_e, half1)
24        self.t_e_3_head2 = nn.Linear(in_e, half2)
25
26
27        self.t_g_head1 = nn.Linear(in_g, half1)
28        self.t_g_head2 = nn.Linear(in_g, half2)
29
30
31        self.o3 = nn.Linear(self.nb_triplet_fts, self.out_size)
32
33
34        self.aggr_mixer = nn.Linear(2 * self.nb_triplet_fts, self.nb_triplet_fts)
35        self.res_gate = nn.Linear(in_e + in_g, 1)
36
37
38        self.gate_node = nn.Linear(in_z, 6)
39        self.gate_edge = nn.Linear(in_e, 6)
40        self.gate_global = nn.Linear(in_g, 2)
41
42    def _compute_tri_msgs(
```

```
43        self,
44        z: _Array,
45        edge_fts: _Array,
46        graph_fts: _Array,
47    ) -> Optional[_Array]:
48
49
50
51      if not self.use_triplets:
52        return None
53
54
55
56      gate_node_all = torch.sigmoid(self.gate_node(z))
57      gate_edge_all = torch.sigmoid(self.gate_edge(edge_fts))
58      gate_global = torch.sigmoid(self.gate_global(graph_fts))
59
60
61
62      gate_t1 = gate_node_all[..., 0:2]
63      gate_t2 = gate_node_all[..., 2:4]
64      gate_t3 = gate_node_all[..., 4:6]
65
66      tri_1 = torch.cat([
67          self.t_1_head1(z) * gate_t1[..., 0:1],
68          self.t_1_head2(z) * gate_t1[..., 1:2]
69      ], dim=-1)
70      tri_2 = torch.cat([
71          self.t_2_head1(z) * gate_t2[..., 0:1],
72          self.t_2_head2(z) * gate_t2[..., 1:2]
73      ], dim=-1)
74      tri_3 = torch.cat([
75          self.t_3_head1(z) * gate_t3[..., 0:1],
76          self.t_3_head2(z) * gate_t3[..., 1:2]
77      ], dim=-1)
78
79
80      gate_te1 = gate_edge_all[..., 0:2]
81      gate_te2 = gate_edge_all[..., 2:4]
82      gate_te3 = gate_edge_all[..., 4:6]
83
84      tri_e_1 = torch.cat([
85          self.t_e_1_head1(edge_fts) * gate_te1[..., 0:1],
86          self.t_e_1_head2(edge_fts) * gate_te1[..., 1:2]
87      ], dim=-1)
88      tri_e_2 = torch.cat([
89          self.t_e_2_head1(edge_fts) * gate_te2[..., 0:1],
90          self.t_e_2_head2(edge_fts) * gate_te2[..., 1:2]
91      ], dim=-1)
92      tri_e_3 = torch.cat([
93          self.t_e_3_head1(edge_fts) * gate_te3[..., 0:1],
94          self.t_e_3_head2(edge_fts) * gate_te3[..., 1:2]
95      ], dim=-1)
96
97
98      tri_g = torch.cat([
99          self.t_g_head1(graph_fts) * gate_global[:, 0:1],
100         self.t_g_head2(graph_fts) * gate_global[:, 1:2]
101     ], dim=-1)
102
103     triplets = (
104         tri_1.unsqueeze(2).unsqueeze(3) +
105         tri_2.unsqueeze(1).unsqueeze(3) +
106         tri_3.unsqueeze(1).unsqueeze(2) +
107         tri_e_1.unsqueeze(3) +
```

```
108              tri_e_2.unsqueeze(2) +
109              tri_e_3.unsqueeze(1) +
110              tri_g.unsqueeze(1).unsqueeze(2).unsqueeze(3)
111          )
112
113
114      max_aggr = torch.amax(triplets, dim=3)
115      mean_aggr = torch.mean(triplets, dim=3)
116      combined = torch.cat([max_aggr, mean_aggr], dim=-1)
117      mixed_aggr = self.aggr_mixer(combined)
118      tri_msgs_main = self.o3(mixed_aggr)
119
120
121      graph_fts_expanded = graph_fts.unsqueeze(1).unsqueeze(2).expand(
122          -1, edge_fts.shape[1], edge_fts.shape[2], -1)
123      gate_input = torch.cat([edge_fts, graph_fts_expanded], dim=-1)
124      gate = torch.sigmoid(self.res_gate(gate_input))
125      residual = self.o3(F.relu(tri_e_1)) * gate
126      tri_msgs = tri_msgs_main + residual
127
128      if self.activation is not None:
129        tri_msgs = self.activation(tri_msgs)
130
131
132      return tri_msgs
```

