# Structured Progressive Knowledge Activation for LLM-Driven Neural Architecture Search

## Abstract

This paper focuses on a key challenge in Neural Architecture Search (NAS): integrating established architectural knowledge while exploring new designs under expensive evaluations. Large language models (LLMs) are a promising assistant for NAS because they can translate rich architectural and coding priors into executable code edits. However, in practice, seemingly local revisions often propagate into non-local behavioral and performance shifts because a single edit can inadvertently couple multiple interacting functional factors, a phenomenon we refer to as functional entanglement. To make LLM knowledge usable under such entanglement, we propose Structured Progressive Knowledge Activation (SPARK), which activates relevant priors by explicitly selecting the functional factor to modify and conditioning the edit on that factor. This factor-conditioned editing reduces entangled side effects and yields more targeted, reliable architecture modifications. On CLRS-DFS, SPARK achieves a $28.1\times$ sample-efficient architecture evolution speedup and yields a 22.9% relative improvement in OOD accuracy.

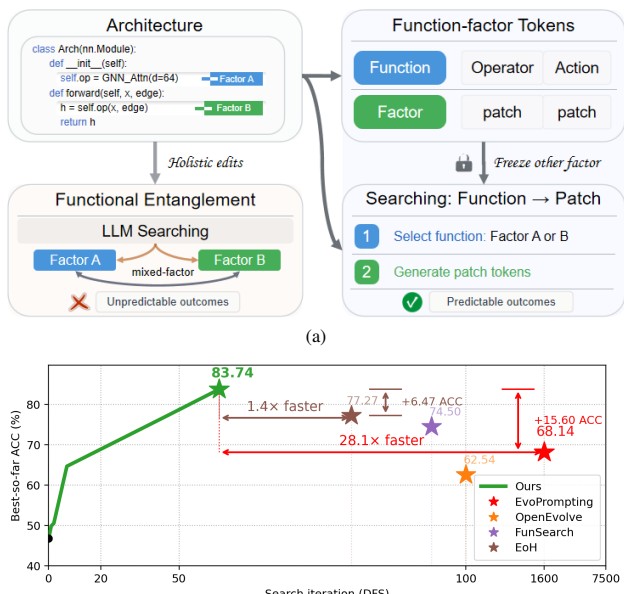

*Figure 1.* Motivation for structure-guided editing in LLM-driven NAS. (a) Free-form edits entangle functional factors and often break interfaces, while factor-scoped tokens enable where-to-how edits to isolate the intervention target. (b) CLRS-DFS results show faster progress and higher OOD accuracy with fewer search iterations.

## 1. Introduction

This paper studies a central tension in Neural Architecture Search (NAS): integrating established architectural knowledge while exploring new designs under expensive evaluations. This tension is exacerbated by the enormous search space of neural architectures and the high cost of training and evaluation, which makes naive exploration impractical (Zoph & Le, 2017b; Real et al., 2017; Liu et al., 2019). LLM-driven NAS is a promising direction because it brings

[1]Anonymous Institution, Anonymous City, Anonymous Region, Anonymous Country. Correspondence to: Anonymous Author <anon.email@domain.com>.

Preliminary work. Under review by the International Conference on Machine Learning (ICML). Do not distribute.

this balance into executable code space: a language model can propose or revise architecture programs by drawing on rich coding and architectural priors, while feedback from training and evaluation guides subsequent iterations (Chen et al., 2023). Yet, in practice, these priors often fail to translate into reliable and controllable edits, becoming a key bottleneck for effective search.

A primary reason is that architecture programs encode multiple interacting functional decisions whose dependencies are largely implicit in code. As illustrated in Fig. 1a (left), "Holistic edits" in free-form "LLM Searching" often update "Factor A" and "Factor B" together (a "mixed-factor" revision); we call this failure mode functional entanglement. In our setting, "Factor A" corresponds to the Operator factor (which computation module is used), and "Factor B" corresponds to the Action factor (how the operator is invoked and

wired under interface and shape constraints). Both factors are functional because together they specify the executable computation graph that determines model behavior. When a single revision entangles these factors, even seemingly local code changes can propagate into non-local behavior and performance shifts and can also break executability (e.g., by violating interface or shape consistency), yielding "Unpredictable outcomes."

These observations suggest that the bottleneck in LLM-driven NAS is not the absence of architectural knowledge, but the lack of a mechanism that turns LLM priors into controllable edits under functional entanglement. To make priors actionable, revisions should be factor-isolated under functional entanglement, matching the constraint in Fig. 1a (right) ("Freeze other factor"). We therefore introduce "Function-factor Tokens" that expose functional factors as explicit, selectable choices, instantiated as Operator and Action, each paired with a dedicated patch slot for localized code changes. This representation supports the structured editing primitive "Searching: Function→Patch": first "Select function: Factor A or B" (i.e., select Operator or Action), then "Generate patch tokens" to realize an executable update conditioned on the selected factor, which promotes more "Predictable outcomes" by isolating the edited factor.

Guided by this principle, we propose Structured Progressive Knowledge Activation (SPARK), a structure-guided editing operator for LLM-driven NAS. SPARK turns free-form rewriting into "Searching: Function→Patch" by decomposing each revision into the two explicit steps in Fig. 1a: "Select function: Factor A or B" and "Generate patch tokens," while enforcing "Freeze other factor" through factor-conditioned patching. By making the intervention factor explicit and restricting each step to one factor at a time, SPARK reduces cross-factor interference between Operator and Action, mitigates entangled side effects, and improves the executability and reliability of generated candidates.

We evaluate SPARK on the CLRS algorithmic reasoning benchmark (Veličković et al., 2022), which instantiates NAS as program-structured architecture tasks under a standardized training and evaluation protocol. On CLRS-DFS, SPARK achieves a 28.1× sample-efficient architecture evolution speedup and improves OOD accuracy from 68.14% to 83.74% (a +15.6-point gain) (Fig. 1b). Across 10 CLRS tasks, SPARK achieves 83.92% mean OOD accuracy with essentially unchanged compute relative to

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

: proposal attempts vs. evaluations.** We distinguish proposal attempts (LLM edit calls; indexed by iteration) from evaluations. A candidate counts as one evaluation only if it is executable and passes feasibility checks, and is then fully trained and scored under the CLRS protocol. Non-executable proposals are rejected and therefore do not count as evaluations. Accordingly, our reliability curves (Fig. 2) report metrics over a fixed 100-attempt trajectory, while our evaluation-efficiency statement (consistent with the Abstract/Introduction) is reported in terms of the number of evaluated candidates.

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

**Evaluation protocol (DFS search; reporting on 10 tasks).** For each method, we run an LLM-driven evolutionary search on the CLRS `dfs` task for 100 proposal attempts under the same search budget and evaluation harness, and measure OOD accuracy using the CLRS OOD split. We then take the incumbent best architecture discovered on `dfs` and train/evaluate it from scratch on an additional 9 representative CLRS tasks. We report (i) per-task OOD accuracy and MACs on these 10 tasks (Table 1), and (ii) reliability/progress curves over the 100-attempt trajectory (Fig. 2). (Full results over all CLRS tasks are provided in the appendix.)

**Baselines and reproducibility.** We compare against EvoPrompting (Chen et al., 2023), OpenEvolve (Sharma, 2025), FunSearch (Romera-Paredes et al., 2024), and EoH (Liu et al., 2024) (as listed in Table 1). Since the OpenEvolve

and FunSearch papers do not explicitly report results on CLRS, we obtain their CLRS

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

The supplementary material reports per-task results on all 18 CLRS tasks, including full accuracy tables beyond the main text. It also includes multi-seed experiments and complete evolution traces (the discovered architectures and best-so-far trajectories across iterations) to further support the reliability and sample-efficiency claims.

## 5. Conclusion

We study LLM-driven neural architecture search as multi-round evolution over executable programs and identify cross-factor functional entanglement as a key reliability bottleneck. We propose SPARK, which enables factor-isolated editing via factor selection and factor-conditioned patch generation, instantiated on CLRS with Operator/Action factors and feasibility checks.

On CLRS, SPARK improves over the reference on 12/18 tasks and outperforms EvoPrompting on 10/18, reaching 83.74% OOD on DFS with 57 evaluated candidates for a 28.1× speedup under our evaluation accounting.

## 6. Impact Statement

This paper presents work whose goal is to advance the reliability of LLM-driven architecture evolution and neural architecture search. While the techniques studied here may improve the efficiency and reproducibility of model development, we do not anticipate direct negative societal impacts beyond those commonly associated with general-purpose machine learning research.

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

# A. supplementary for Structured Progressive Knowledge Activation for LLM-Driven Neural Architecture Search

## A.1. Overall Comparison with EvoPrompting on CLRS

Table A.1 reports the overall per-task performance comparison between our method and EvoPrompting on the CLRS benchmark.

**Search dynamics under structure-guided evolution.** Fig. A.1 characterizes the search behavior of our Structured Progressive Knowledge Activation (SPARK) over the 100 generated candidates on CLRS, aggregated across five random seeds. Recall that SPARK factorizes each evolution step into where-to-edit and how-to-refine decisions: the Architecture Scope Router (ASR) selects an explicit scope (e.g., operator-level vs. action), the Refinement Compass (RC) specifies scope-local refinement directives, and the Scoped Architecture Refiner (SAR) applies edits only within the chosen scope under feasible-set refinement constraints (syntax/interface/shape feasibility), preventing cross-factor entanglement.

**Progress (best-so-far OOD accuracy).** In Fig. A.1a, the best-so-far OOD accuracy rises sharply in the early phase and then shows a saturation trend, indicating that SPARK can convert a small candidate budget into substantial performance gains, while later iterations provide more incremental improvements. The thick mean curve together with the min–max band suggests this pattern is consistent across seeds rather than being driven by a single favorable run. We attribute the strong early gains to SPARK's structured (scope-local) updates: by committing each iteration to an explicit scope via ASR and applying RC-guided, SAR-executed edits within a constrained feasible set, the search produces composable improvements that accumulate across rounds instead of being washed out by unconstrained rewrites.

**Cost control (MACs of the incumbent best model).** Fig. A.1b tracks the MACs of the incumbent best-so-far model throughout the same 100-attempt trajectory. Although MACs may change when the incumbent is replaced, the curves do not exhibit a systematic upward trend and remain within a stable range across seeds. This indicates that the accuracy gains in Fig. A.1a are not primarily explained by progressively increasing compute, but rather by discovering more effective structures within a comparable compute envelope—consistent with SPARK's goal of structure-guided refinement rather than cost-increasing scaling.

**Why both plots.** We report both progress and cost because reliable multi-round LLM-driven NAS requires improvements to be (i) attributable and composable across iterations (reflected by best-so-far accuracy gains) while also being

| Task | CLRS | EvoPrompting | Ours |
|------|------|--------------|------|
| DFS | 46.78 | 68.14 | 83.74 |
| BFS | 88.32 | 99.99 | 70.56 |
| FW | 48.95 | 61.43 | 32.91 |
| AP | 88.32 | 93.46 | 97.90 |
| TSch | 87.25 | 88.23 | 87.44 |
| QS | 0.47 | 0.79 | 10.69 |
| CC | 43.43 | 41.86 | 77.34 |
| ASel | 95.18 | 95.05 | 98.03 |
| BFord | 97.39 | 97.50 | 96.04 |
| DSP | 98.19 | 98.01 | 97.80 |
| FMS | 76.36 | 75.35 | 85.64 |
| Dij | 96.05 | 97.30 | 99.22 |
| LCS | 80.51 | 85.75 | 87.60 |
| MCO | 91.68 | 90.77 | 87.29 |
| Min | 97.78 | 98.40 | 99.07 |
| MST | 86.39 | 88.74 | 88.67 |
| SI | 97.64 | 98.15 | 97.07 |
| Topo | 87.27 | 88.12 | 99.95 |

*Table A.1.* Per-task OOD accuracy (%) on CLRS. Abbreviations: FW=Floyd–Warshall, AP=Articulation Points, TSch=Task Scheduling, QS=Quickselect, CC=Connected Components, ASel=Activity Selector, BFord=Bellman–Ford, DSP=DAG Shortest Paths, FMS=Find Maximum Subarray (Kadane), Dij=Dijkstra, LCS=LCS Length, MCO=Matrix Chain Order, Min=Minimum, MST=MST Prim, SI=Segments Intersect, Topo=Topological Sort.

(ii) cost-bounded (reflected by stable MACs). Together, Fig. A.1a and Fig. A.1b provide complementary evidence that SPARK achieves effective, budget-efficient progress without relying on an expanding computational footprint.

## A.2. Program Evolution Analysis Across Iterations

This section analyzes how the PGN triplet module evolves across iterations. We treat `_maybe_lazy_init` as the Operator (parameterization/structure definition) and `_compute_tri_msgs` as the Action (triplet message computation). Iteration 0 is the base program we aim to evolve (Listing 1); later iterations introduce increasingly expressive, stable, and structure-aware computation.

**Iteration 0 (Base): additive triplet construction with max aggregation.** As shown in Listing 1, the base Operator instantiates linear projections for node features (`z`), edge features (`edge_fts`), and graph-level features (`graph_fts`). The base Action constructs a triplet tensor via broadcasting (`unsqueeze`) and combines components primarily through additive composition followed by a max-style reduction. Benefit: this design is simple, stable, and provides a strong baseline with clear gradient paths, serving as

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

```