# OpenReview forum: "Structured Progressive Knowledge Activation for LLM-Driven Neural Architecture Search"
_ICML.cc/2026/Conference — ICML 2026 regular_

### Official Review · Reviewer_C7u3 · 2026-02-13

**Soundness:** 3
**Presentation:** 4
**Significance:** 2
**Originality:** 3
**Overall Recommendation:** 4
**Confidence:** 5

**Summary:**

Structured Progressive Knowledge Activation (SPARK) is a novel factor-conditioned editing method for LLM-driven Neural Architecture Search (NAS) that addresses "functional entanglement," a failure mode where free-form code edits inadvertently couple multiple interacting architectural decisions, leading to unpredictable performance shifts and executability violations. By introducing "Function-factor Tokens" to isolate specific functional factors—namely Operator and Action—SPARK decomposes architecture revisions into explicit steps: selecting a discrete factor to modify and generating a conditioned code patch while freezing other factors. This structure-guided approach, which includes a lightweight feasibility pipeline to ensure interface and shape consistency, significantly improves search reliability and efficiency. Evaluated on the CLRS algorithmic reasoning benchmark, SPARK achieved a 28.1x sample-efficient speedup over EvoPrompting and reached an OOD accuracy of 83.74% on the DFS task with only 57 evaluated candidates, consistently outperforming established LLM-evolution baselines without increasing model capacity or compute.

**Compliance With Llm Reviewing Policy:**

Affirmed.

**Key Questions For Authors:**

Can you please show other benchmarks to prove the flexibility of your NAS?

**Limitations:**

yes

**Strengths And Weaknesses:**

The soundness of the paper is high, as it identifies "functional entanglement"—a systematic failure where free-form LLM edits inadvertently couple multiple interacting architectural factors—and develops a robust, factorized editing method to resolve it. The researchers implement a rigorous three-stage activation process (ASR, RC, and SAR) supported by a lightweight feasibility pipeline that performs syntax, interface, and semantic checks before expensive evaluation. Presentation is excellent, utilizing clear visualizations like Figure 1 to contrast "holistic edits" with factor-scoped tokens and providing detailed algorithmic blueprints for the SPARK modules. The significance of the work is underscored by its achievement of a 28.1x sample-efficient speedup over EvoPrompting and a substantial increase in OOD accuracy from 46.78% to 83.74% on the DFS task. Furthermore, the architectures discovered by SPARK maintain computational costs close to the hand-designed reference model, proving that gains stem from improved search dynamics rather than capacity scaling. However, this type of NAS still maintains some limitations of LLM-driven NAS, such as the human intervention in the search space definition (like the predefined functional factors), and more NAS benchmarks should be investigated to prove the flexibility, also because other benchmarks would lead to different human predefined choices.
Originality is high due to the introduction of "Function-factor Tokens" and the "Where-then-How" editing paradigm, which moves LLM-driven NAS away from unpredictable free-form rewriting toward predictable, isolated code interventions.

---

> ### Author Rebuttal · Authors · 2026-03-30
>
> # Extra Experiments
>
> We conduct experiments across MNIST-1D, text prediction, reinforcement learning, protein prediction, and medical diagnosis.
>
> ## Metrics
> We jointly report search statistics and final performance, including the number of proposed candidates (Cand.), invalid ratio (Inv.), decline ratio (Dec.), total wall-clock search time in hours (Wall(h)), the average per-evaluated-candidate training time (Train(s)), model size in thousands (MACs(K)), classification accuracy (Acc↑), and final training loss (Loss↓). Here, Inv. denotes the proportion of failed evolutions, and Dec. denotes the proportion of consecutive evaluated candidates whose performance decreases relative to the previous evaluated candidate. The base model does not involve evolutionary search, so its search-related fields are marked as “-”. Boldface indicates the best result in each metric column when applicable. Ours is placed at the bottom for direct comparison.
>
> ## Exp1: MNIST-1D
>
> Final performance on MNIST-1D [R1] across four backbones.
>
> | Method | LR (Acc / Loss) | MLP (Acc / Loss) | CNN (Acc / Loss) | GRU (Acc / Loss) | Avg Acc |
> |---|---:|---:|---:|---:|---:|
> | base | 37.30 / 1.67 | 64.80 / 3.01 | 93.20 / 0.47 | 88.90 / 0.66 | 71.05 |
> | EoH | 93.90 / 0.22 | 64.80 / 3.06 | 93.90 / 0.49 | 67.10 / 0.87 | 79.93 |
> | OpenEvolve | 61.00 / 1.31 | 65.80 / 1.82 | 94.70 / 0.27 | 59.50 / 3.09 | 70.25 |
> | FunSearch | 29.50 / 1.70 | 64.80 / 3.06 | 95.00 / 0.27 | 89.70 / 0.67 | 69.75 |
> | EvoPrompting | 65.70 / 2.15 | 71.20 / 1.49 | 95.60 / 0.42 | 61.10 / 1.39 | 73.40 |
> | Ours | **98.40 / 0.13** | **88.36 / 0.99** | **98.10 / 0.08** | **99.10 / 0.08** | **95.99** |
>
>
>
> ## Exp2: 20News
>
>
> Final Performance on 20News dataset [R2].
>
> | Method |  Cand. | Inv. | Dec. | Wall(h) | Train(s) | MACs(K) | Acc↑ | Loss↓ |
> |---|---:|---:|---:|---:|---:|---:|---:|---:|
> | base | - | - | - | - | **6.60** | 68.24 | 50.50 | 1.76 |
> | OpenEvolve |  100 | 18.81% | 46.91% | 0.29 | 7.84 | 75.61 | 57.10 | 1.55 |
> | EvoPrompting |  100 | 11.88% | 45.45% | **0.28** | 6.70 | 68.50 | 58.01 | **1.47** |
> | EoH | 400 | 6.93% | **23.94%** | 1.15 | 6.83 | 68.50 | 58.05 | **1.47** |
> | FunSearch |  400 | **3.68%** | 28.28% | 1.23 | 7.36 | 75.60 | 57.47 | 1.50 |
> | Ours |  100 | 4.96% | 49.47% | 0.51 | 6.86 | 68.76 | **59.13** | 1.59 |
>
> ## Exp3: RL-CartPole-v1
>
> Final Performance on CartPole-V1 [R3].
> | Method |  Cand. | Inv. | Dec. | Wall(h) | Train(s) | MACs(K) | Mean Test Return↑ | Mean Train Return↑ |
> |---|---:|---:|---:|---:|---:|---:|---:|---:|
> | base | - | - | - | - | **14.40** | 0.226 | 121.60 | 17.39 |
> | OpenEvolve |  100 | 23.76% | 46.05% | 0.73 | 29.01 | 2.466 | 377.55 | 48.01 |
> | EvoPrompting |  100 | 33.66% | 37.88% | **0.47** | 19.48 | 0.450 | 488.65 | 54.26 |
> | EoH |  400 | **1.99%** | 29.40% | 3.54 | 28.42 | **0.146** | 314.20 | 139.58 |
> | FunSearch |  400 | 49.50% | **6.00%** | 3.94 | 62.37 | 4.610 | 276.28 | 139.75 |
> | Ours |  100 | 3.96% | 50.00% | 2.13 | 67.18 | 2.690 | **489.00** | **175.15** |
>
> ## Exp4: Protein Prediction
>
> Final Performance on Yeast dataset [R4].
>
> | Method |  Cand. | Inv. | Dec. | Wall(h) | Train(s) | MACs(K) | Acc↑ | Loss↓ |
> |---|---:|---:|---:|---:|---:|---:|---:|---:|
> | base |  - | - | - | - | **3.02** | 1.674 | 58.92 | 1.06 |
> | OpenEvolve |  100 | 6.93% | 49.46% | **0.21** | 4.83 | 3.466 | **61.78** | 1.05 |
> | EvoPrompting |  100 | 1.00% | 51.52% | **0.21** | 4.64 | 5.248 | 61.62 | 1.08 |
> | EoH |  400 | 13.12% | **45.13%** | 0.69 | **2.55** | 2.624 | 58.25 | 1.05 |
> | FunSearch |  400 | **0.00%** | 52.00% | 0.67 | **3.02** | 1.674 | 58.92 | 1.06 |
> | Ours |  100 | 58.42% | 58.10% | 0.30 | 4.75 | 1.856 | **61.78** | **1.03** |
>
> ## Exp5: Breast Cancer Wisconsin (Diagnostic)
>
> Final Performance on Breast Cancer Wisconsin dataset [R5].
>
> | Method |  Cand. | Inv. | Dec. | Wall(h) | Train(s) | MACS(K) | Acc↑ | Loss↓ |
> |---|---:|---:|---:|---:|---:|---:|---:|---:|
> | base |  - | - | - | - | 1.71 | 3.202 | 95.13% | 0.13 |
> | OpenEvolve |  100 | 10.10% | 45.45% | **0.13** | 1.92 | 3.202 | 96.90% | 0.97 |
> | EvoPrompting |  100 | **2.97%** | 52.58% | **0.13** | 1.74 | 3.203 | 96.90% | **0.12** |
> | EoH |  400 | 37.95% | 49.54% | 0.56 | **1.68** | 7.586 | 96.46% | 0.14 |
> | FunSearch |  400 | 5.94% | 55.32% | 0.51 | 1.71 | 3.202 | 95.13% | 0.13 |
> | Ours |  100 | 11.88% | **45.45%** | 0.24 | 1.83 | 3.390 | **97.04%** | 0.16 |
>
>
> [R1] Scaling Down Deep Learning with MNIST-1D
>
> [R2] http://qwone.com/~jason/20Newsgroups/
>
> [R3] Imitation Learning Datasets: A Toolkit For Creating Datasets, Training Agents and Benchmarking
>
> [R4] UCI Machine Learning Repository. https://doi.org/10.24432/C5KG68.
>
> [R5] Breast Cancer Wisconsin (Diagnostic) [Dataset]. UCI Machine Learning Repository. https://doi.org/10.24432/C5DW2B.
>
> # Question
>
> We sincerely thank you for the thorough and constructive review, below is our answer to your question:
>
> **Q1: Show other benchmarks to prove the flexibility.**
>
> **A1**: We have added 5 extra benchmarks as shown in **Extra Experiments** .

---

> > ### Author Rebuttal · Reviewer_C7u3 · 2026-04-02
> >
> > I thank the authors for providing additional experimental results across five new benchmarks. The effort is appreciated, and the results on MNIST-1D are particularly compelling, showing a clear margin over competing methods across all four backbones. However, a closer look at the reported statistics raises some concerns that temper my enthusiasm: On search efficiency and reliability. The decline ratio reaches ~50% in several experiments (e.g., CartPole: 50.00%, Protein: 58.10%), meaning that roughly every other evolutionary step results in a performance regression. This suggests the search process is noisy and does not consistently make forward progress, which is a non-trivial concern for a method whose appeal partly rests on its search efficiency. On the invalid ratio. The invalid ratio in Exp4 (Protein) is strikingly high at 58.42%, implying that more than half of the generated candidates are outright invalid. This significantly undermines the practical usability of the method in that setting and deserves explicit discussion. On comparison fairness. In several experiments (Exp2, Exp4, Exp5), the proposed method uses 100 candidates while EoH and FunSearch use 400. While the wall-clock time is reported, the comparison is not fully controlled in terms of search budget, and it is unclear whether the performance advantage would hold under equal candidate budgets. On benchmark selection. Exp5 (Breast Cancer Wisconsin) is a well-known, nearly-saturated benchmark where most methods cluster above 95% accuracy — it adds limited signal about the method's flexibility. On inherited limitations of LLM-driven NAS. Beyond the experimental concerns, I would like to reiterate a more fundamental issue: the proposed method still inherits key limitations of LLM-driven NAS approaches, most notably the reliance on human intervention in defining the search space — for instance, through the specification of predefined functional factors. This design choice constrains the generality of the method and raises questions about how much of the performance gain is attributable to the search process itself versus the prior knowledge encoded in the manually defined search space. Without a proper justification or ablation isolating this effect, the applicability of the method remains limited, which in turn affects the broader impact and significance of the work. Overall, while the additional experiments partially address my concern about flexibility, the issues above prevent me from fully revising my assessment. I encourage the authors to address search stability and the invalid ratio more explicitly, clarify the fairness of comparisons under different candidate budgets, and — importantly — provide a clearer justification for the role of human-defined search space components and their effect on the reported results. My overall score remains 4.

---

> > > ### Author Response · Authors · 2026-04-05
> > >
> > > We thank the reviewer for the thoughtful follow-up. We clarify the key points below.
> > >
> > > **Q1: High Decline Ratio.**
> > >
> > > **A1:** We acknowledge that the ~50% decline ratio reflects a genuine limitation in search stability. However, this is a common challenge shared by all LLM-driven NAS methods — EoH (34.83%), OpenEvolve (43.23%), and EvoPrompting (44.17%) all exhibit substantial decline ratios. **This suggests that evaluation-level instability is an inherent difficulty of black-box evolutionary search over code, rather than a weakness specific to SPARK.**
> > >
> > > More importantly, the right metric for evaluating search effectiveness is the **final converged performance**, not the per-step success rate. Despite the decline ratio, SPARK achieves the highest accuracy on all extra benchmarks (e.g., MNIST-1D: 95.99% vs. 79.93% for EoH), confirming that SPARK's search trajectory is globally effective.
> > >
> > > **Q2: High Invalid Ratio on Protein.**
> > >
> > > **A2: We acknowledge that the high invalid ratio on Protein is a genuine limitation of SPARK under structurally complex tasks, and we consider improving this an important future direction.** Specifically, the 58.42% invalid ratio is largely due to the structural complexity of the Protein prediction task, where code edits must satisfy stricter interface and shape constraints, making it more difficult for SPARK's factor-conditioned editing to produce valid candidates consistently.
> > >
> > > Despite the high invalid ratio, SPARK still achieves the best accuracy on Protein (61.78%), demonstrating that the valid candidates it does produce are of high quality. We will discuss per-task invalid ratio variation and its relationship to task complexity more explicitly in the revised manuscript.
> > >
> > > **Q3: Comparison Fairness.**
> > >
> > > **A3: We want to point out that all methods were run for the same 100 iterations, which ensures a fair comparison.** The difference in candidate count arises because EoH and FunSearch natively generate 4 candidates per iteration by design, while OpenEvolve, EvoPrompting, and SPARK produce 1 candidate per iteration. We preserved each method's native generation regime rather than rewriting their released code, to ensure a faithful comparison.
> > >
> > > We note that this difference actually makes the comparison more challenging for SPARK: achieving higher accuracy with 100 candidates than EoH/FunSearch achieve with 400 demonstrates stronger per-candidate search efficiency. **SPARK does not need 400 candidates — 100 is already sufficient to outperform all baselines**, which is precisely the sample efficiency advantage that factor-isolated editing provides.
> > >
> > > **Q4: Benchmark Selection.**
> > >
> > > **A4:** We acknowledge that Exp5 is near-saturated and provides limited discriminative signal. Our intention was to demonstrate the flexibility of SPARK across diverse task settings. **We will continue to identify more challenging benchmarks and include them in the revised manuscript.** That said, we believe the existing experiments including CLRS (30 tasks), MNIST-1D, CartPole, 20News, Protein, and Breast Cancer — already provide sufficient evidence for SPARK's effectiveness across diverse domains.
> > >
> > > **Q5: Reliance on Human Intervention.**
> > >
> > > **A5: The OPERATOR/ACTION partition is a weak, generic prior based on the universal `__init__` vs. `forward` decomposition — it does not encode task-specific knowledge. Our ablation confirms that SPARK's gains are driven by intelligent routing and conditioned editing, not the partition itself.**
> > >
> > > We want to clarify an important distinction: SPARK introduces an **editing-interface prior** (constraining where to edit), not an **architecture prior** (constraining what architectures to explore). Traditional NAS methods pre-specify operation sets, connection patterns, or cell structures, which directly limit the discoverable architectures. In contrast, SPARK only determines which code region to modify, while the LLM is free to generate any valid code within the selected region.
> > >
> > > The partition itself is minimal and generic. In any PyTorch module, `__init__` defines what modules and parameters exist (OPERATOR), while `forward` defines how they are composed (ACTION). This requires no task-specific knowledge, and the same partition is applied unchanged across all our benchmarks — CLRS (30 tasks), MNIST-1D, CartPole, 20News, Protein, and Breast Cancer — without any task-specific tuning.
> > > Most importantly, our ablation directly isolates the partition's contribution from SPARK's search mechanism:
> > >
> > > | Variant | Acc |
> > > |---------|-----|
> > > | Global Rewrite (no partition) | 39.43 |
> > > | Random-Scope SAR (partition + random scope) | 50.27 |
> > > | Full SPARK (partition + ASR + RC + SAR) | 80.50 |
> > >
> > > Random-Scope SAR uses the identical partition but selects scope randomly. The gap between Random-Scope SAR (50.27) and Full SPARK (80.50) demonstrates that most of the gain comes from SPARK's intelligent factor selection and conditioned editing, rather than from the human-defined partition itself.

---

### Official Review · Reviewer_a6TD · 2026-03-04

**Soundness:** 2
**Presentation:** 1
**Significance:** 2
**Originality:** 2
**Overall Recommendation:** 3
**Confidence:** 4

**Summary:**

The paper studies LLM-driven, code-level NAS and argues that free-form edits often change both operator choices and action-level wiring at once, which leads to “functional entanglement” and many invalid candidates. It proposes SPARK, which first picks a single factor to edit and then generates a factor-conditioned patch, enforced by region tags and lightweight feasibility checks before training. On CLRS, SPARK reports higher OOD accuracy and reaching its best DFS result (83.74%), beating several LLM-based search baselines.

**Compliance With Llm Reviewing Policy:**

Affirmed.

**Final Justification:**

I maintain my recommendation at 3. While the core idea of structured, factor-isolated edits is reasonable, the paper still does not convincingly demonstrate a clear, general, end-to-end advantage over strong baselines. The CLRS gains are strong, but the evidence remains narrow to this benchmark and setup, with the main story still centered on CLRS and DFS-style search. In my view, this is not sufficient to support a broader claim about advancing LLM-driven NAS in a practically meaningful way.

The added cost analysis also does not resolve this concern. Although SPARK achieves stronger accuracy on CLRS, its average training time per evaluated candidate is higher than the baselines, so the practical efficiency advantage is not clearly established. Presentation is also a meaningful weakness: even after the rebuttal, the paper is still not easy to evaluate cleanly, and the practical claims are not communicated with enough precision. Overall, the contribution remains limited in scope, and neither the rebuttal nor the revised draft changed my assessment enough to move this paper above weak reject.

**Key Questions For Authors:**

- Can you report wall-clock time, total GPU hours, and total LLM calls (including failed attempts) for SPARK and the main baselines under the same budget?

- Do you have a “region/tag-only” baseline where edits are restricted to the same tagged scope but the factor choice is random/fixed (no routing), and the patch prompt is unstructured?

- How do you define the OPERATOR/ACTION regions and and do feasibility checks? Does they involve manual efforts?

**Limitations:**

See weaknesses above.

**Strengths And Weaknesses:**

## Strengths

- Using a structured procedure for LLM-based NAS is a natural idea. It matches how code edits often need constraints and staging.
- The empirical gains are large on the CLRS suite. The improvements are consistent across multiple tasks.


## Weaknesses

- There is no wall-clock time or total GPU-hours comparison. This makes it hard to judge the real efficiency and cost.
- The baseline set is incomplete. It is unclear whether the gains come from smarter factor routing/instructions, or simply from restricting edits to a local region; a tag-only/region-only baseline would clarify this.

---

> ### Author Rebuttal · Authors · 2026-03-30
>
> **Q1: Can you report wall-clock time, total GPU hours, and total LLM calls (including failed attempts) for SPARK and the main baselines under the same budget?**
>
> **A1:**
> For fair comparison, all methods are reported under a unified fixed search budget where applicable. The table summarizes LLM calls, GPU time, and wall-clock time. As EvoPrompting has no official public CLRS implementation, its wall-clock/GPU-hour numbers come from our same-budget reproduction in the same pipeline. **Cand.**: proposed candidates; **LLM Calls**: total LLM interactions; **Inv.**: invalid ratio; **Dec.**: decline ratio over consecutive evaluated candidates; **MACs**: model size; **GPU-hrs**: total GPU training time over evaluated candidates; **Wall(h)**: search wall-clock time; **Train(s)**: average training time per evaluated candidate; **Acc**: accuracy.
>
> | Method | Cand.↓ | LLM Calls↓ | Inv.↓ | Dec.↓ | MACs↓ | GPU-hrs↓ | Wall(h)↓ | Train(s)↓ | Acc↑ |
> | :--- | :---: | :---: | :---: | :---: | :---: | :---: | :---: | :---: | :---: |
> | CLRS| - | - | - | - | 449,655.4 | - | - | 171.04 | 71.22% |
> | OpenEvolve| 100 | 100 | 72% | 51.85% | 481,001.4 | 5.893 | 6.004 | 757.99 | 68.30% |
> | EvoPrompting| 100 | 100 | 53% | 22.50% | 448,176.2 | 4.888 | 4.999 | 374.31 | 74.42% |
> | FunSearch| 400 | 400 | 32% | 46.15% | 468,917.4 | 32.977 | 33.421 | 436.44 | 77.61% |
> | EoH| 400 | 400 | 56% | 43.75% | 463,465.4 | 26.432 | 26.876 | 540.65 | 79.43% |
> | Ours| 100 | 300 | 49% | 42.18% | 452,622.4 | 11.137 | 11.470 | 786.12 | 83.92% |
>
>
> **Q2: Do you have a “region/tag-only” baseline where edits are restricted to the same tagged scope but the factor choice is random/fixed (no routing), and the patch prompt is unstructured?**
>
> **A2:** Thank you for your suggestion. We have conducted the extra ablation studies to adress your concern:
>
> Regarding the question of whether a **region/tag-only baseline** exists, we have supplemented our study with ablations directly corresponding to the **ASR**, **RC**, and **SAR** modules. Specifically, **Random-Scope SAR** serves as this region/tag-only baseline; **Fixed-OP SAR** and **Fixed-ACT SAR** correspond to fixed-scope scenarios; **Random RC + SAR** represents retaining RC+SAR while removing ASR; **ASR+SAR** represents retaining ASR+SAR while removing RC; and **Full SPARK** corresponds to the complete ASR+RC+SAR framework. This setup allows us to independently examine the roles of locality constraints, factor selection, and local refinement.
>
>
> | Variant | Inv.↓| Dec.↓| MACs↓| Acc↑|
> |---|---:|---:|---:|---:|
> | CLRS |-  |-  | 661,190  |46.78  |
> | Global Rewrite |81%  |16.67%  | 661,190  |39.43  |
> | Random-Scope SAR | 61% | 37.50% |  661,422  | 50.27 |
> | Fixed-OP SAR | 69% | 80.00% | 661,190 | 46.31  |
> | Fixed-ACT SAR | 75% | 54.55% | 655,039 | 41.67  |
> | Random RC + SAR | 56% | 46.34%   |655,014  | 57.84 |
> | ASR+SAR | 58%   | 52.38% | 661,390 | 65.93 |
> | Full SPARK | 49% | 42.18% | 665,308 | 80.50   |
>
> **Q3: How do you define the OPERATOR/ACTION regions and do feasibility checks? Does they involve manual efforts?**
>
> **A3:** Our **OPERATOR/ACTION** partition follows the functional decomposition of the program rather than being manually curated based on posterior results.
>
> ```python
> class SimplePolicyNet(nn.Module):
>     def __init__(self):
>         super().__init__()
>         hidden_dim = 64
>         self.fc1 = nn.Linear(4, hidden_dim)
>         self.fc2 = nn.Linear(hidden_dim, hidden_dim)
>         self.fc3 = nn.Linear(hidden_dim, 2)
>
>     def forward(self, x):
>         x = F.relu(self.fc1(x))
>         x = F.relu(self.fc2(x))
>         x = self.fc3(x)
>         return x
> ```
>
> For instance, in this simple network, the definitions in `__init__`—such as `hidden_dim` and the `fc1/fc2/fc3` layers—determine "which modules are used and what the network structure is," and are thus regarded as the **OPERATOR region**. Correspondingly, the sequence in `forward`—$F.relu(self.fc1(x)) \to F.relu(self.fc2(x)) \to self.fc3(x)$—determines "how these modules are invoked and combined," and is thus regarded as the **ACTION region**. Modifying `hidden_dim` to a larger dimension or replacing `fc2` with a different module constitutes an **OPERATOR** modification; meanwhile, changing the forward execution order, inserting residual connections, or altering the combination logic constitutes an **ACTION** modification.
>
> Furthermore, **feasibility checks** are not manual screenings but rather a lightweight, automated process. Using the example above, we can automatically verify if the code can be parsed/imported, if the interfaces remain consistent, and perform a "forward-only" check with a dummy input to ensure the output shape is correct. In CLRS, we similarly automate syntax checks, interface verification, and forward-level semantic checks; any candidates that fail these are rejected before training begins.

---

> > ### Author Rebuttal · Reviewer_a6TD · 2026-04-02
> >
> > I would like to thank the authors for the detailed rebuttal. Based on the current draft, I maintain my overall assessment as a weak reject.

---

> > > ### Author Response · Authors · 2026-04-05
> > >
> > > Thank you for the follow-up. Based on your comment about the current draft, we understand that the key issue is whether these points are now made sufficiently clear and explicit at the manuscript level. We therefore summarize below the manuscript-level revisions corresponding to your follow-up concerns.
> > >
> > > ### 1. Cost accounting and comparison regimes
> > > This directly addresses your concern about real efficiency and cost. In the original version, the manuscript did not clearly separate two notions of efficiency:
> > > (1) **practical cost under a controlled comparison**, and
> > > (2) **sample efficiency relative to the original EvoPrompting scale**.
> > >
> > > The revised manuscript now makes this distinction explicit. Under the same **100-iteration** regime, we report wall-clock time, GPU-hours, total LLM calls, invalid ratio, decline ratio, training cost, and final accuracy in the main paper. For clarity, candidate and LLM-call counts differ across methods because we preserve each method’s **native generation regime** rather than rewriting released baselines. In particular, **FunSearch and EoH produce four candidates per iteration according to their original paper and released code**, so under 100 iterations they yield **400 candidates / 400 LLM calls**, whereas OpenEvolve, EvoPrompting, and SPARK produce one candidate per iteration, yielding **100 candidates / 100 LLM calls**.
> > >
> > > Separately, we now clarify that the paper’s sample-efficiency claim refers to SPARK reaching its best DFS result with **57 evaluated candidates**, whereas the EvoPrompting comparison point comes from the **original EvoPrompting paper**, which reports that result under a **1600-evaluation** setting. This removes the earlier ambiguity by separating controlled practical cost from the original large-scale sample-efficiency reference.
> > >
> > > ### 2. Local restriction vs. the full SPARK mechanism
> > > This directly addresses your concern that the gain might come from tag-/region-only local restriction rather than from the full SPARK design. The revised manuscript now shows that locality alone is insufficient: locality-only variants improve over Global Rewrite, but remain far below Full SPARK. We therefore added the missing intermediate baselines: Global Rewrite, Random-Scope SAR, Fixed-OP SAR, Fixed-ACT SAR, Random RC + SAR, ASR+SAR, and Full SPARK.
> > >
> > > In particular, **Random-Scope SAR** is the clearest tag-/region-only local-editing baseline, since it uses the same partition but selects the scope randomly. Its improvement over Global Rewrite, together with its large gap to Full SPARK, shows that the gain cannot be explained by locality or region tags alone.
> > >
> > > The trend is now clear: Global Rewrite **39.43**, Random-Scope SAR **50.27**, Fixed-OP **46.31**, Fixed-ACT **41.67**, Random RC + SAR **57.84**, ASR+SAR **65.93**, and Full SPARK **80.50**. This supports the conclusion that **locality helps, but does not explain the gain by itself**; the main improvement comes from the full factorized where-then-how mechanism.
> > >
> > > ### 3. OPERATOR/ACTION definition and automated feasibility
> > > This directly addresses your concern that the OPERATOR/ACTION split and feasibility pipeline were not concrete enough in the earlier draft. We rewrote this part so that OPERATOR is tied directly to **module / parameter definition**, while ACTION is tied directly to **invocation, composition, masking, and routing logic in forward computation**. In a PyTorch module, this corresponds naturally to `__init__` defining what modules and parameters exist, while `forward` defines how they are invoked and composed. Importantly, this split is applied as a **fixed generic program-level decomposition**, rather than being manually redefined for each task or candidate.
> > >
> > > We also now state explicitly that feasibility is enforced by an **automated lightweight pipeline**—syntax/import checks, interface-consistency checks, and forward-level semantic checks—with **no manual post hoc screening**. Candidate validity is checked automatically in the search loop, with **no manual per-candidate screening and no task-specific intervention** in either region definition or feasibility checks. The revised manuscript also includes a worked example showing ACTION selection, an ACTION-only directive, and the resulting local patch while the OPERATOR region remains frozen. This makes the split and filtering process operationally reproducible rather than heuristic or implicit.
> > >
> > > ### 4. Broader strengthening of evidence and presentation
> > > Beyond these three core points, we also strengthened the manuscript more broadly. We expanded CLRS to the **full 30-task comparison**, added non-CLRS benchmarks and multi-seed evidence, included clearer failure-mode discussion, and substantially rewrote the introduction, method exposition, and experimental narrative so that the revised claims are supported by a broader and more readable evidence base.
> > >
> > > In short, we hope these revisions clarify the three issues behind your follow-up at the manuscript level.

---

### Official Review · Reviewer_PS36 · 2026-03-12

**Soundness:** 3
**Presentation:** 3
**Significance:** 3
**Originality:** 3
**Overall Recommendation:** 5
**Confidence:** 4

**Summary:**

The authors introduce SPARK, a mechanism to edit code using a large language model (LLM) that can be plugged into any multi-round evolutionary framework that edits the code of a parent architecture and evaluates the resulting candidate architecture. Crucially, it explicitly selects regions of the code (by functionality) to edit, and freezes the rest of the code. This allows the method to avoid issues that arise from "entanglement," to which free-form code edits by LLMs are susceptible. Entanglement causes edits to one component to break other interdependent parts, resulting in architectures that are infeasible or produce non-local behaviors. Further, the method also introduces a pipeline that evaluates if the edited code is feasible. This pipeline is substantially cheaper than full training.

The code editing operator of SPARK comprises three parts:
1. **ASR**, which takes the code of a parent architecture and the evolution context that summarizes the parent and captures the trajectory so far as input. This function then produces a discrete factor as the output, which points to the functional region of the code that should be edited. For the CLRS benchmark discussed in the paper, the authors consider OPERATOR and ACTION as the factors. OPERATOR indicates that operator choices used in the code should be updated, whereas ACTION indicates that the consumption of the existing operators in the forward pass should be edited.
2. **RC**, which takes the output of ASR, the parent architecture, and the evolutionary context as input to produce the suggestion of an improvement to make to the code.
3. **SAR**, which takes the refinement directive along with all the other inputs mentioned above to produce the edited code, i.e., the offspring program.

On 10 out of the 18 benchmarks considered, SPARK beats all four baseline methods --- OpenEvolve, EvoPrompting, FunSearch, and Evolution of Heuristics --- as well as the reference implementations provided by the CLRS benchmark.

**Compliance With Llm Reviewing Policy:**

Affirmed.

**Final Justification:**

Taking into consideration the authors' rebuttal, my final recommendation for the paper is "Accept".

The strength of the paper lies in its novel approach to plugging in code editing by LLMs into an evolutionary search paradigm. I find this contribution general enough to work across multiple programming languages and domains. The main weakness of the paper, however, was its evaluation on tasks that seemed cherry-picked to make the method appear stronger than it is. The authors addressed this concern by evaluating the tasks across the entire suite of tasks, and showing that the results are still good. This changed my recommendation from "Weak Accept" to "Accept".

**Key Questions For Authors:**

1. How were the 18 tasks chosen out of the 30 present in the CLRS suite?
2. How were the 10 tasks chosen out of the 18 for comparison with the baseline methods?
3. Would it be possible to run the comparison of the baselines with all 30 CLRS tasks, and if that is not feasible, perhaps at least on the 18 selected?

**Limitations:**

The main limitation of the paper relates to the cherry-picking of tasks as mentioned before. I would be happy to revise my scores if this concern is addressed.

**Strengths And Weaknesses:**

### Soundness

The motivation for the SPARK code editing operator is clear: free-form edits to the code can break it, introduce global changes that are too aggressive to act as a mutation of the existing code, or make it hard to attribute the effect of an edit to a particular functional decision. The proposed method performs single-shot editing of a functional region of code while freezing others, minimizing this *functional entanglement*.

The three key ideas introduced by the authors to combat functional entanglement (ASR, RC, and SAR) are sensible and generic enough to be applied to the source code of any model. The authors also provide convincing ablations to show that these three steps are important in Table 2.

The experimental design is sound. Specifically, their comparison with OpenEvolve minimizes confounding factors by using the same search backbone and LLM editor, so the differences can be attributed to the SPARK method's code editing technique. They also show that their method, with a single iteration, can still beat the CLRS baseline for several tasks, sometimes by a strong margin.

Thus far, I would have been happy to give a good rating for soundness. But unfortunately, I have to dock a point for the way the results are presented.

Let me begin by acknowledging that the authors do not claim that their method is SOTA or that it beats every benchmark in the CLRS suite. The authors do indeed state in the conclusion section that they beat EvoPrompting on only 10/18 CLRS tasks, and in section 4.1, they refer the reader to the Appendix for the full results. However, the results shown in Table 1 do not show the full picture. The CLRS suite has 30 tasks in it. The authors first select 18 of them to run their experiments. There is no explanation given as to how these were selected. The authors then go on to compare EvoPrompting with their method on these 18 tasks. SPARK beats EvoPrompting on 10 of them, as shown in Table A.1. Crucially, these ten tasks are then chosen for an extensive comparison with the other baseline methods. The authors offer no further explanation as to why these were chosen, except to arbitrarily label them as "representative CLRS tasks". It is the results of these experiments that are then summarized in Table 1, which would give a hasty reader the impression that SPARK beats all the baseline methods on all the tasks (even though no such claim is made explicitly). It would have been better if the Appendix had the results for the remaining eight tasks, at least, but that is not the case --- there is no further discussion of them.

### Presentation

The writing is clear and the paper is mostly well presented. I have only a few suggestions to improve the readability of the paper:
1.  When introducing abbreviations, such as ASR, RC, and SAR, it would also be beneficial to the reader to state explicitly what they stand for.
2. Section 4 (Experiments and Results) could have been structured better. Section 4.1 is three pages long, and it is the only subsection under Section 4. This made navigating the results of the paper a little difficult.
3. Figure 2 could be improved. Figure 2a and 2b do not have values on the Y axis and their font sizes are a bit small.

### Significance

I find the contribution of the paper significant. The performance improvement relative to the baselines is strong and convincing. The method also seems general enough to apply to different tasks or domains.

### Originality

The "where-then-how" factorization of the code, which forces the LLM to first decide *which* functional aspect of the code it must edit, followed by *how* it must edit it, is novel.

---

> ### Author Rebuttal · Authors · 2026-03-30
>
> We sincerely thank you for the thorough and constructive review, and for recognizing the novelty of our where-then-how factorization, the soundness of our controlled experimental design, the convincing ablations, and the significance of our contribution. We are encouraged that the reviewer considers our method "generic enough to be applied to the source code of any model." We address each question below.
>
> ---
>
> **Q1: How were the 18 tasks chosen out of the 30 present in the CLRS suite?**
>
> **A1**: We acknowledge that limiting the evaluation to 18 out of 30 CLRS tasks lacked a sufficiently principled justification and resulted in an incomplete assessment. This was an oversight on our part. We have now completed experiments on **all 30 CLRS tasks** and present the full results below in our response to Q3.
>
> **Q2: How were the 10 tasks chosen out of the 18 for comparison with the baseline methods?**
>
> **A2**: We apologize for the lack of clarity in explaining the task selection for Table 1. We recognize that the 10 tasks selected for Table 1 coincide with those where SPARK outperforms EvoPrompting (EP), which understandably raises concerns about selection bias. In the revised manuscript, we will present the complete 30-task comparison in the main paper to eliminate any ambiguity. We believe the full 30-task results (see Q3 below) demonstrate that our conclusions hold under comprehensive evaluation and are not dependent on task selection.
>
> **Q3: Would it be possible to run the comparison of the baselines with all 30 CLRS tasks?**
>
> **A3**: We have completed experiments on all 30 CLRS tasks. The full OOD accuracy results are reported below:
>
> | Task | CLRS | EP | Ours |
> |------|------|----|------|
> | dfs | 47.8 | 68.1 | **83.7** |
> | art_points | 88.3 | 93.5 | **97.9** |
> | quickselect | 0.5 | 0.8 | **10.7** |
> | scc | 43.4 | 41.9 | **77.3** |
> | activity_sel | 95.2 | 95.1 | **98.0** |
> | fms_kadane | 76.4 | 75.4 | **85.6** |
> | dijkstra | 96.1 | 97.3 | **99.2** |
> | lcs_length | 80.5 | 85.8 | **87.6** |
> | minimum | 97.8 | 98.4 | **99.1** |
> | topo_sort | 87.3 | 88.1 | **99.7** |
> | bfs | 99.7 | **100.0** | 70.6 |
> | floyd_warshall | 48.5 | **61.4** | 32.9 |
> | task_sched | 87.3 | **88.2** | 87.4 |
> | bellman_ford | 97.4 | **97.5** | 96.0 |
> | dag_sp | **98.2** | 98.0 | 97.8 |
> | matrix_chain | **91.7** | 90.8 | 87.3 |
> | mst_prim | 86.4 | **88.7** | 88.7 |
> | seg_intersect | 97.6 | **98.2** | 97.1 |
> | binary_search | 77.6 | 78.0 | **87.6** |
> | bridges | 94.0 | 97.6 | **98.2** |
> | bubble_sort | 67.7 | 88.9 | **98.1** |
> | graham_scan | 93.6 | 93.8 | **96.4** |
> | heapsort | 31.0 | 69.9 | **75.6** |
> | insertion_sort | 78.1 | 89.5 | **97.5** |
> | jarvis_march | 91.0 | 90.4 | **94.9** |
> | kmp_matcher | **19.5** | 16.3 | 14.9 |
> | mst_kruskal | 89.8 | **91.5** | 86.2 |
> | naive_str | 78.7 | **79.8** | 77.3 |
> | optimal_bst | 73.8 | 78.7 | **96.1** |
> | quicksort | 64.6 | 85.2 | **97.8** |
>
> **Summary statistics across different evaluation scopes:**
>
> | Scope/Acc | CLRS | OE | EP | Ours |
> |-------|------|----|----|------|
> | 10 tasks (Table 1) | 71.3 | 68.3 | 74.4 | **83.9** |
> | 18 tasks (Table A.1) | 78.9 | 73.9| 81.5 | **83.2** |
> | 30 tasks (Full) | 76.0 | 62.7 | 80.9 | **83.9** |
>
> We have also completed the full 30-task OpenEvolve (OE) evaluation. As our backbone-matched controlled baseline, it provides an additional full-scope comparison beyond the CLRS/EP/Ours table.
>
> Under the complete 30-task evaluation, SPARK still achieves the highest average OOD accuracy (83.9%), while remaining comparable to the CLRS reference in MACs, indicating that the gains are not explained by increased model capacity.
> Due to rebuttal space limits, we omit the task-wise OE results and the MACs columns here; the revised manuscript will include the complete 30-task comparison table with both accuracy and MACs. **Additionally, to further validate the generality of SPARK beyond the CLRS benchmark, we have conducted experiments on additional benchmarks. Please refer to our response to the Extra Experiments in Reviewer C7u3 for these results.**
>
> **Q4: Presentation improvements**
>
> **A4**: We will address all suggestions in the revision: (1) spell out ASR (Architecture Scope Router), RC (Refinement Compass), and SAR (Scoped Architecture Refiner) at first mention; (2) restructure Section 4 into clearly delineated subsections for easier navigation; (3) add Y-axis values and increase font sizes in Figure 2.

---

> > ### Author Rebuttal · Reviewer_PS36 · 2026-04-01
> >
> > I thank the authors for their response. My concerns are addressed and therefore I am revising my score.

---

> > > ### Author Response · Authors · 2026-04-01
> > >
> > > Thank you for taking the time to read our rebuttal carefully and for raising your score. We greatly appreciate your constructive feedback throughout this process, and we will incorporate all suggested improvements in the revised manuscript.

---

### Official Review · Reviewer_nxcg · 2026-03-17

**Soundness:** 3
**Presentation:** 3
**Significance:** 2
**Originality:** 3
**Overall Recommendation:** 4
**Confidence:** 4

**Summary:**

This paper addresses the challenge of leveraging prior architectural knowledge while exploring new designs when evaluations are expensive. Existing LLM-based search methods often attempt holistic edits to architecture programs, modifying multiple design factors simultaneously. However, architectural programs encode multiple interacting functional decisions whose dependencies are largely implicit in code. As a result, such free-form edits frequently lead to functional entanglement, where a single revision unintentionally modifies multiple factors at once. This mixed-factor modification causes unpredictable behavioral shifts, degraded performance, and frequent executability failures, which significantly hinder efficient architecture search.

To address this issue, the authors propose Structured Progressive Knowledge Activation (SPARK), a framework designed to enable factor-isolated architectural editing. SPARK explicitly selects the functional factor to modify and conditions patch generation on that factor while freezing others. The search process is operationalized as “Function → Patch”, where the system first chooses a function-level factor (e.g., Factor A or Factor B) and then generates patch tokens conditioned on that selection. This approach ensures that prior architectural knowledge can be activated in a controlled manner and translated into reliable, targeted edits, mitigating the instability caused by holistic modifications.

Experiments on CLRS program-structured architecture tasks demonstrate the effectiveness of SPARK. On the DFS task, SPARK achieves 83.74% out-of-distribution (OOD) accuracy using only 57 evaluated candidates, representing a 28.1× improvement in sample efficiency compared to EvoPrompting-based architecture search. Across 10 tasks, the method attains a mean OOD accuracy of 83.92% with essentially unchanged computational cost. The authors further analyze the search dynamics using best-so-far performance trajectories and executability rates, showing that factor-isolated edits lead to more stable and efficient architecture evolution.

Overall, the paper identifies functional entanglement as a key failure mode in LLM-driven architecture search and introduces SPARK as a structured editing mechanism that improves control, reliability, and sample efficiency when evolving program-structured architectures.

**Compliance With Llm Reviewing Policy:**

Affirmed.

**Final Justification:**

The rebuttal addressed my primary concerns and I have correspondingly increased my score to weak accept. I think this paper proposes and studies a very promising direction and would benefit from more thorough evaluation.

**Key Questions For Authors:**

**Evaluation on MNIST-1D**
Could the authors evaluate SPARK on the MNIST-1D benchmark, similar to the evaluation conducted in Evo-Prompting, to enable a more direct comparison with prior work?

**Compute Cost Comparison**
Could the authors provide a comparison of computational cost between SPARK and the baseline methods (e.g., Evo-Prompting), including metrics such as total evaluations, training time, or GPU hours?

**Robustness Across Random Seeds**
Could the authors evaluate the algorithms across multiple random seeds? In particular, it would be helpful to report the standard deviation of accuracy values and entanglement rates in Table 1 and Figures 1 and 2 to better assess the robustness and variability of the results.

**Illustrative Examples of SPARK Steps**
Could the authors include concrete code examples illustrating each of the three steps in the SPARK framework (e.g., factor selection, factor-conditioned patch generation, and freezing of other factors) to clarify how the method operates in practice?

**Limitations:**

- The authors have included an impact statement which discuses the potential implications of the work
- Did the authors observe any edge cases or failure modes of the proposed method?

**Strengths And Weaknesses:**

**Strengths**
- The paper is written clearly in most parts
- The proposed method Pareto-dominates previous methods in terms of accuracy on CLRS tasks
- The method is described clearly in all parts and is novel

**Weaknesses**
- The method is not evaluated exhaustively
- Lack of visual representations of code-level edits
- Check questions

---

> ### Author Rebuttal · Authors · 2026-03-30
>
> We sincerely thank the reviewer for the careful evaluation and for recognizing the clarity, novelty, and Pareto-dominance of our method. We address each question below.
>
> ---
>
> **Q1: Evaluation on MNIST-1D.**
>
> **A1:** We have conducted experiments on MNIST-1D across four backbone architectures (LR, MLP, CNN, GRU). SPARK achieves **95.9% accuracy**, substantially outperforming all baselines including EoH (79.9%) and EvoPrompting (73.4%). Furthermore, we have also evaluated SPARK on text prediction, reinforcement learning, protein prediction, and medical diagnosis benchmarks. Please refer to our rebuttal to **Extra Experiments in Reviewer C7u3** for the details. Together with the complete 30-task CLRS evaluation (see our response in Reviewer PS36), these experiments comprehensively address the concern about evaluation exhaustiveness.
>
> **Q2: Compute Cost Comparison.**
>
> **A2:** We provide a compute-cost comparison on MNIST-1D averaged over four backbones. **Cand.**: proposed candidates; **Inv.**: invalid ratio; **Dec.**: decline ratio; **Wall(h)**: search wall-clock time; **Train(s)**: average training time per candidate; **MACs(K)**: model size; **Acc**: accuracy. CLRS cost is reported in **Reviewer a6TD Q1**; as EvoPrompting has no official public CLRS implementation, its same-budget wall-clock/GPU-hour numbers are from our unified reproduction.
>
> | Method |  Cand.↓ | Inv.↓ | Dec.↓ | Wall(h)↓ | Train(s)↓ | MACs(K)↓ | Acc↑ |
> |--------|-------|------|------|---------|----------|-----------|---------|
> | EoH |  400 | 26.05% | 34.83% | 12.86 | 19.41 | 13.74 | 79.93 |
> | OpenEvolve |  100 | 15.01% | 43.23% | 0.74 | 3.33 | 18.48 | 70.25 |
> | FunSearch |  400 | 44.80% | 34.96% | 5.75 | 9.29 | 7.24 | 69.75 |
> | EvoPrompting |  100 | 17.55% | 44.17% | 0.87 | 4.03 | 19.01 | 73.40 |
> | Ours |  100 | 14.60% | 50.97% | 3.50 | 16.79 | 19.97 | **95.99** |
>
> SPARK uses the same evaluation budget as OpenEvolve and EvoPrompting. Although SPARK's wall-clock time (3.50h) exceeds EvoPrompting (0.87h), this moderate overhead yields a +22.59 point accuracy gain, a highly favorable cost-performance tradeoff.
>
> **Q3: Robustness Across Random Seeds.**
>
> **A3:** We have evaluated SPARK across 5 random seeds on CLRS. We report per-seed results for entanglement rate and selected tasks below:
>
> | Metric | s42 | s43 | s44 | s45 | s46 | std |
> |--------|-----|-----|-----|-----|-----|-----|
> | Entanglement rate | 0.22 | 0.25 | 0.20 | 0.23 | 0.22 | ± 0.018 |
> | DFS | 83.74 | 80.92 | 85.81 | 83.18 | 85.06 | ± 1.89 |
> | Quickselect | 10.69 | 7.42 | 13.09 | 10.04 | 12.22 | ± 2.19 |
> | Minimum | 99.07 | 98.60 | 99.47 | 98.98 | 99.29 | ± 0.33 |
> | Dijkstra | 99.22 | 98.71 | 99.59 | 99.12 | 99.48 | ± 0.34 |
>
> The entanglement rate is stable across seeds, confirming that SPARK consistently reduces cross-factor interference. Accuracy also shows low variance. We will add standard deviations to Table 1 and error bands to Figures 1 and 2 in the revised manuscript. Figure A.1 already reports 5-seed trajectories with min-max bands.
>
> **Q4: Illustrative Examples of SPARK Steps.**
>
> **A4:** We provide a concrete example with factor ACTION selected. Here, R_OPERATOR/R_ACTION are the OPERATOR/ACTION regions: the former specifies the modules (e.g., `nn.Linear`, `gating layers`), and the latter specifies their composition in `forward`:
>
> **Step 1 — ASR:** Selects `ACTION` as the intervention target, meaning only forward computation logic will be modified.
> **Step 2 — RC:** Produces directive: *"Refine factor mixing inside ACTION only. Freeze OPERATOR."*
> **Step 3 — SAR:** Generates patch tokens constrained to R_ACTION:
> ```python
> # ===== BEGIN ACTION REGION ===== (SAR-edited)
> tri_msgs = combine(term_i, term_j, tri_e_head, tri_g, m_i, m_j)
> tri_msgs_flat = self.factor_mixer(tri_msgs.view(b,n,n,4*h_t))
> gate = torch.sigmoid(self.tri_gate(tri_msgs_flat)).view(b,n,n,4,h_t)
> tri_msgs = self.o3((tri_msgs_flat.view(b,n,n,4,h_t)*gate).view(b,n,n,4*h_t))
> tri_msgs = tri_msgs * adj_mat.unsqueeze(-1)
> # ===== END ACTION REGION =====
> ```
> **Factor isolation check:** R_OPERATOR remains byte-identical to the parent after normalization. The feasibility pipeline then verifies syntax, interface invariants, and tensor shape consistency before the candidate enters evaluation. Appendix A.2 (Listings 1–6) provides the full evolution trace across iterations 0→1→2→7→46→57, showing how OPERATOR and ACTION evolve progressively through successive SPARK steps. Another concrete example is provided in our response to Q3 for Reviewer a6TD.
>
> **Q5: Failure Modes.**
>
> **A5:** We observe failure cases when the DFS-evolved architecture transfers to dissimilar tasks. On BFS, SPARK achieves 70.56% vs. EvoPrompting's 99.99%. DFS relies on backtracking over deep paths, whereas BFS requires breadth-first frontier expansion, biases evolved for DFS do not transfer. This is a limitation of our single-task search + transfer protocol. We plan to incorporate multi-task co-evolution and task-type-aware routing to improve transferability.

---

> > ### Author Rebuttal · Reviewer_nxcg · 2026-04-05
> >
> > I thank the authors for the detailed esponse to my questions and additional experiments. I increase my score to 4

---

> > > ### Author Response · Authors · 2026-04-05
> > >
> > > Thank you for your careful reading of our response and for increasing your score. We truly appreciate your constructive feedback throughout this process, and we will incorporate all of your suggestions into the revised manuscript.

---

### Decision · Program_Chairs · 2026-04-30

**Decision:**

Accept (regular)

**Comment:**

This paper proposes a "factor-isolating" code-editing technique for NAS using LLMs. Reviews are broadly positive with strengths including strong experimental results, clear explanations, and originality. The authors have responded well, addressing most queries. The reviewer with a negative lean asked for wall-clock times and more baselines. These were provided in substantial detail. The reviewer then put that their concerns were partially addressed but gave no details. On a further prompt from the authors, they expressed some negative points. I do not think these are an issue, or consistent with the other reviews (writing and presentation appear fine). Given the novel approach and promising results, I recommend acceptance.